# VISOGENDER: A dataset for benchmarking gender bias in image-text pronoun resolution

**Siobhan Mackenzie Hall**      **Fernanda Gonçalves Abrantes**      **Hanwen Zhu**

**Grace Sodunke**      **Aleksandar Shtedritski**      **Hannah Rose Kirk**

Oxford Artificial Intelligence Society, University of Oxford

## Abstract

We introduce VISOGENDER, a novel dataset for benchmarking gender bias in vision-language models. We focus on occupation-related biases within a hegemonic system of binary gender, inspired by Winograd and Winogender schemas, where each image is associated with a caption containing a pronoun relationship of subjects and objects in the scene. VISOGENDER is balanced by gender representation in professional roles, supporting bias evaluation in two ways: i) *resolution bias*, where we evaluate the difference between pronoun resolution accuracies for image subjects with gender presentations perceived as masculine versus feminine by human annotators and ii) *retrieval bias*, where we compare ratios of professionals perceived to have masculine and feminine gender presentations retrieved for a gender-neutral search query. We benchmark several state-of-the-art vision-language models and find that they demonstrate bias in resolving binary gender in complex scenes. While the direction and magnitude of gender bias depends on the task and the model being evaluated, captioning models are generally less biased than Vision-Language Encoders. Dataset and code are available at https://github.com/oxai/visogender.

## 1  Introduction

Vision-language models (VLMs) are advancing rapidly and reaching ever-wider audience across numerous applications, such as classification and captioning, as well as text-to-image retrieval and generation. However, these models are pre-trained from uncurated image-text pairs scraped from the internet [1, 2] and so, their outputs can perpetuate or amplify social biases [3, 4, 5, 6]. How the VLM is used determines the mechanisms of how biases transfer from pre-training to downstream representational and/or allocational harms [7]. For example, a VLM used for image retrieval may skew towards returning more images of male doctors, thus reinforcing stereotypical associations between gender and career success; or a VLM used for captioning may more frequently misgender women or non-binary image subjects, aligning with a capability (un)fairness [8].

Despite the growing body of work on evaluating and mitigating the bias of VLMs [9, 10, 11, 12, 13], there is a dearth of specifically-designed benchmark datasets to evaluate the presence of social biases across downstream tasks (such as captioning or image retrieval): most prior work has measured biases using pre-existing image datasets such as FairFace [14] or COCO [15, 16], despite their limited real-world transferability and spurious correlations [12, 17]. In this paper, we introduce the VISOGENDER benchmark for evaluating bias in VLMs. The design of VISOGENDER is inspired by two prior bodies of works. Firstly, we apply stress-testing of vision-linguistic reasoning capabilities of VLMs as in the Winoground benchmark [18] but introduce the dimension of social biases. Secondly, we adopt

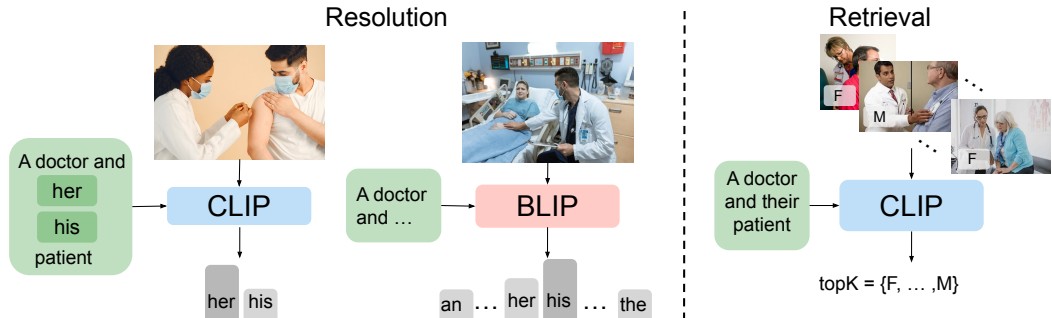

Figure 1: **Resolution of gender pronouns and retrieval with a neutral query.** We resolve gender by (i) using zero-shot classification with Cross Models Encoders, such as CLIP, and (ii) next-token prediction with captioning models, such as BLIP. We have an additional simpler task to resolve the gender of a single person, e.g., with a template "The doctor and her / his stethoscope".

the templated structure to test gender bias in occupational pronoun resolution from NLP research, but apply it to the vision-language domain—specifically the WinoGender [19] and WinoBias [20] frameworks, in turn inspired by Winograd Schema [21]. To our knowledge, VISOGENDER is the first dataset to combine both of these contributions by stress-testing *gender bias* in visual-linguistic reasoning and coreference resolution capabilities.

VISOGENDER contains images of a person depicted in an occupational role ("the doctor"), combined with either an object ("the stethoscope") or a participant ("the patient"). Each image is labelled by human annotators for the perceived gender presentation of the occupation and/or the participant, and the dataset is balanced across different genders occupying these roles. For each image, we construct natural language captions that use the perceived gender labels of the image subject(s) to derive a possessive binary pronoun relationship ("the doctor and his/her patient"). We test bias in two tasks: pronoun *resolution* and image *retrieval* (see Fig. 1 for summary). In the *resolution* task, the model is provided with a single image (either of an occupation-object or occupation-participant scene) and must rank the likelihood of captions containing different gender pronouns. There are varying levels of difficulty in the resolution task—from a single person resolution in the occupation-object case; to two person resolution in the occupation-participant case, where either both subjects are perceived to have the same gender presentation (easier), or different gender presentations (harder). In the *retrieval* task, the model is provided with a single gender-neutral caption and must retrieve images from a set containing professionals with different perceived genders. We measure *resolution bias* using the gender accuracy gap in correct pronoun resolution (corresponding to capability fairness) and *retrieval bias* using commonly-applied metrics such as Bias@K, MaxSkew and NDKL (corresponding to representational fairness).

We present preliminary results for six state-of-the-art Vision-Language Encoders (VLEs) models [1, 22, 2, 23, 24, 25] and two state-of-the-art captioning models [26, 27]. We find that models still struggle to resolve pronoun relationships, especially when there are two people in the image of different perceived gender presentations (where performance is close to random). Our benchmark also recovers that (i) models display substantial accuracy gaps in resolving pronouns of masculine-versus feminine-presenting subjects, indicating the presence of resolution bias; and (ii) when provided with a neutral pronoun query ("the doctor and *their* patient"), models predominately rank images of masculine-presenting subjects higher than those with a perceived feminine presentation, indicating a retrieval bias. We compare these results to U.S. Labor Force statistics (as in [28, 20, 19]) and find some correlations between model bias and societal occupational skew in the US labour market. Our findings demonstrate there is still substantial progress to be made in improving scene disambiguation for visual-linguistic reasoning, as well as reducing the gender gap in resolution performance and retrieval outcomes. Some caveats to our findings are needed: first, while we do find capability unfairness evidenced in differential performance across gender identity groups, this paper works within a hegemonic system of binary and stereotypical gender presentation that remains prevalent in Western constructions and perceptions of gender, where datasets typically originate [29, 30]. Models are likely to perform even more poorly in downstream tasks where there is additional complexity introduced by the inclusion of greater cultural diversity, as well as transgender, non-binary and gender-diverse individuals who are underrepresented here and in other vision-language datasets.

Second, our benchmark is designed to measure poor performance on the two tasks, thus identifying potential downstream harms from occupational gender bias in VLMs. However, being good at the task requires not only advanced visual-linguistic reasoning but also accurate gender prediction, a capability that can be concerning if misused for surveillance purposes. We discuss both of these limitations and concerns in Sec. 5. The pace at which VLMs are being developed is only set to grow in coming years—VISOGENDER provides a much-needed benchmark to evaluate their potential downstream harms before large-scale deployment.

## 2 Related Works

**Bias in coreferences in NLP** Coreference resolution aims to identify which mentions in a natural language document link to the same real-world entity [31]. In the past decade, significant progress has been made moving from rule-based systems and expert knowledge [32], to statistical models [33, 34] and deep neural networks [35, 36, 37, 38, 39]. Pronoun resolution involves linking a noun such as "doctor" to a pronoun in the sentence. Biases have been identified, with respect to machine translation [40], non-binary pronouns [41], and favouring masculine entities when resolving gender ambiguous cases [42]. Our work is most similar to gender pronoun resolution tasks based on Winograd schemas [21], like Winogender [19] and WinoBias [20] which investigate occupational-related biases. Both of these works demarcate "hard" and "easy" cases based on (anti-)stereotypical gender-occupation associations as measured relative to U.S. Labor Force statistics. We extend this work to the vision-language domain. In our resolution task, we modify the typical Winograd scheme because the correct resolution is unambiguous, i.e., there is a correct caption (and pronoun) for a corresponding image. However, our retrieval task is a closer vision-language analogy to [19, 20] because there is no groundtruth for a "correct" ranking of images given a gender-neutral search query.

**Evaluating visual reasoning** There is an emerging body of work on visual reasoning tasks [43], such as VQA [44, 45, 46], visual word sense disambiguation [47], compositionality understanding [48, 49, 50], comprehension [51] or visual entity linking [52]. Most similar to our work, Winoground [18] evaluates visio-linguistic compositional reasoning by tasking a model to match two images with two captions containing the same set of words, only in a different order—such as "there is a mug in some grass" vs. "there is some grass in a mug". The task is challenging, with state-of-the-art VLMs rarely performing better than chance, though [53] demonstrate some of these failures may be due to atypical images in the dataset. Our vision-linguistic stress-tests are inspired by adapting Winoground to social biases, but a key difference is that our caption-image pairs do not contain the exact same set of words—for example, matching "the doctor and her patient" versus "the doctor and his patient".

**Measuring bias in vision-language models** Measuring the social bias of VLMs is a growing area of research. While early works measure misclassification rates into harmful categories [9, 54], more recent methods investigate face-to-text retrieval [11, 10, 55, 56], or captioning [57]. However, these approaches rely on off-the-shelf datasets, such as COCO [16], which have been shown to contain spurious correlations [17] and thus are not suitable for evaluating model bias [12]. Similar to [12], we balance our dataset by gender across different occupational settings, but instead using naturally-occuring images rather than synthetic edits.

## 3 The VISOGENDER Benchmark

The VISOGENDER dataset contains 690 images of people in various occupational settings, where each image is annotated for the perceived gender presentation of the subject(s) in the image. We use these annotations to construct a templated caption of an inferred pronoun relationship. The dataset covers 23 unique occupations in a hierarchical taxonomy. Each occupation appears in the dataset with two template forms—either as a single person in the image with a possessive pronoun to an object ("the doctor and his/her stethoscope"), or as one of two people in the image with a possessive pronoun to a participant ("the doctor and his/her patient") (see Sec. 3.2). A summary of the dataset is presented in Tab. 1. In the following subsections, we introduce the terminology used throughout our paper (Sec. 3.1); describe the dataset (Sec. 3.3); and provide detail of the templates (Sec. 3.2). We then summarise the two types of VLMs which are compatible with VISOGENDER (Sec. 3.4); and finally, define the two tasks through which we measure model bias (Sec. 3.5).

## 3.1 Terminology

- *Perceived gender presentation*: We do not have direct communication with image subjects, so we cannot know the pronoun with which the subjects identify. Instead of referring directly to gender identity, we use perceived gender presentation to indicate that an inference is made by an external human annotator. We do not make any claims of assigning a person's gender identification, which we recognise as an individual's personal experience of gender [58], and acknowledge that our labels may differ from the subject's personal identity.
- *Masculine and feminine presentation*: We opt to use terms for gendered characteristics (masculine and feminine) over terms for biological sex (male and female). These perceived gender labels (denoted as "M" for masculine and "F" for feminine unless otherwise stated), are assigned by an external human annotator based purely on their perception of the person's visual presentation such as facial features, stereotypical clothing, hair or other signals. We recognise that gendered characteristics from masculine to feminine lie on a spectrum and assigning one dominant presentation over another is subject to annotator internal biases, which are in turn endogenous to lived experience. Please see our positionality statement for more information (Sec. 5.3).

## 3.2 Templates

Each templated caption contains three components, adapted from Winogender [19]:

- `Occupation`: a person refered to by an occupational noun and definite article, "the doctor"
- `Pronoun`: a pronoun corresponding to the perceived gender presentation of the `occupation` in the image, e.g., "her" or "his"
- *either* `Object`: a noun corresponding to typical professional items, e.g., "the stethoscope"
- *or* `Participant`: A second person in a typical professional relationship with the `occupation`, e.g., "the patient"

For occupations, we use the list from [19], but remove (i) occupations without a clear possessive pronoun relationship between the occupation and participant, e.g., "the plumber and their houseowner" is not semantically correct; and (ii) occupations without sufficient open-domain images across genders (for both men and women occupying the occupation and participant roles). We classify the remaining occupations into a hierarchical taxonomy to permit aggregate group-wise analysis: **Sector** describes the general field, and includes *education*, *medical*, *office*, *retail* and *service*; **Specialisation** describes subcategories within the sector, where, for example, *services* includes *food services*, *fashion*, *animal* or *household*; and finally, **Occupations** are nested within specialisations, where, for example, *food services* contains *waiter*, *bartender*, and *baker*. Similar to [19, 20, 28], we match U.S. Labor Force statistics on the percentage of men working in each occupation to compare model biases to occupational skew in the real-world US labour market. When comparing our results to these statistics, we will use the perceived masculine gender presentation counts as men ("M"), and the remaining percentages to be for women ("W"). The full taxonomy and list of occupations is presented in the Supplementary Materials. We also source the list of participants from [19] but replace any references to children as participants and in some cases, make modifications for a more natural possessive pronoun, e.g., "the lawyer and the witness" becomes "the lawyer and their client". For objects, we manually define a typical professional item for each occupation. Using these components, we construct three templates (subtasks) of increasing difficulty for coreference resolution:

- **Single subject:** The template of captions is "The {occupation} and {his/her} {object}", e.g., *"the doctor and her stethoscope"*. For each occupation, we collect 10 occupation-object images, 5 for each gender. Here, models only need to resolve the pronoun of one subject in the image, thus testing simple visual-linguistic reasoning.
- **Two subjects of the same perceived gender presentation:** The template of the captions is "The {occupation} and {his/her} {participant}" e.g. *"the doctor and her patient"*. In this case, the perceived gender presentation of the occupation and the participant are the same (both masculine or both feminine). Per occupation, we collect 5 images for each of these two cases (M-M, F-F). Here, the model must resolve the inferred pronouns of two subjects but assigning which subject is the occupation and which is the participant does not affect the prediction.

Table 1: **VISOGENDER dataset summary**, showing the hierarchy of included Sectors, Specialisations, and Occupations; the Perceived Presentation of Gender (P.P.Gender) pairs per template type, and the counts of images within each split of the dataset.

| | Categories | | | | Number of images | | | |
|---|---|---|---|---|---|---|---|---|
| | Sect. | Spec. | Occ. | P.P.Gender Pairs | Images per Occ. | Images per P.P.Gender Pair | Images per P.P.Gender Pair and Occ. | Overall |
| **Single person** (occupation-object) | 5 | 13 | 23 | [M, F] | 10 | 115 | 5 | 230 |
| **Two-person** (occupation-participant) | 5 | 13 | 23 | [MM, FF, MF, FM] | 20 | 115 | 5 | 460 |
| | | | | | | | **Total** | 690 |

- **Two subjects of different perceived gender presentations:** Finally, we use the same occupation-participant template but now the participant and the occupation are of opposite perceived gender presentations (one masculine and one feminine). Per occupation, we collect 5 images for each of case (M-F, F-M). Here, the model must resolve the perceived gender of the subject, *and* infer from image context which is the occupation and which is the participant to infer the pronoun.

### 3.3 Dataset Collection

The VISOGENDER dataset comprises image URLs with annotations for the occupation noun, the participant or object noun, and the perceived gender presentations of the occupation and participant. These annotations can be used to reconstruct the templated captions. Data collection, which includes data labelling, was carried out by the authors of the paper from March to May 2023 on a variety of image databases and search providers, such as Pexels and Google Image Search. We followed a set guidelines to specify exclusion and inclusion criteria, detailed in the Supplementary Materials.[1]

We ensure that there are no duplicate images (no overlaps between occupations) and no invalid URLs across the dataset. In the early stages of data collection, we used the entire list of occupations from [19]. However, we only include those with at least 20 viable URLs (5 per gender pair) for occupation-participants and 10 viable URLs for occupation-object (5 per gender). The image curation process (and availability of viable URLs) is dependent on the retrieval of different gendered roles across occupational search queries and so therefore compromised by inherent representational biases in these search engines. We mitigate effects of imbalance across genders by only including occupations with a full set of images (equal images across all gender pairs) but this may introduce a sample selection bias to the included occupations. Furthermore, inferring gender from an image depends on ingrained biases of the dataset curators. We discuss limitations and biases of data collection in Sec. 5.3, and suggest possible expansions in the future with further resources e.g., partnering with a stock photo company. The dataset is accompanied by Data Clause (which details the Licence and Terms of Use) as well as a Datasheet for Datasets [59] in the Supplementary Materials.

### 3.4 Two Supported Types of Vision-Language Models

VISOGENDER is designed to accommodate two types of VLMs. Here we discuss their properties, and how bias can be measured in common use cases.

**Vision-Language Encoders (VLEs)** VLEs, such as CLIP[1], have separate vision and language encoders and are trained to jointly match images and text. Given an image $i \in \mathcal{R}^{3 \times H \times W}$ and text $t$, a VLE outputs a score $s(i, t)$ that expresses the degree of compatibility between the image and text. The first common use case of VLEs is zero-shot classification of images [1, 60, 61]. This is done by providing a query image $i_q$ and text prompts $t_n, n \in 1, \ldots, N$. For example, if we wish to zero-shot classify the perceived gender of a doctor in an image using pronoun resolution, we can provide text prompts *"This is an image of a doctor and {his, her} notebook"*, and select the pronoun with the highest compatibility score to the image. Such a classifier can be considered biased if, for example, it more accurately infers the pronoun of one perceived gender presentation in some occupations. The

---

[1]For example, images were required to have Creative Commons and/or royalty free licences, and were to be photo-realistic, have maximum two subjects, contain no children nor NSFW content.

second common use case of VLEs is for text-to-image retrieval [1]. Given a text query $t_q$, images $i_n, n \in 1, \ldots, N$ and a query size $K$, we select the $K$ images with the highest compatibility score to the text prompt. In this setting, the model can be biased if, for example, when searching for a given occupation, people from a given demographic are over or under-represented in the top $K$ retrieval results.

**Captioning models** Captioning models are most commonly trained to autoregressively predict a text caption given an image. For an image $i$ and, optionally, a partially completed caption with N tokens $c = [t_1, \ldots t_N]$, the model outputs the probability for the next token $t_{N+1}$ as $p_{\text{cap}}(t_{N+1}|i, t_1, \ldots, t_N)$. Similar to VLEs, we can apply the captioning model to infer the perceived gender of a subject in an image via pronoun resolution. We first supply a query image $i_q$ (say, an image of a doctor) and a caption $c_q$ like *"An image of a doctor and"*. We then inspect the probability distribution for the next token $t_n$, denoted by $p_{\text{cap}}(t_n) = p_{\text{cap}}(t_m|i_q, c_q)$. We can now compare the probabilities $p_{\text{cap}}(t_n) = $ *"her"* and $p_{\text{cap}}(t_n) = $ *"his"*, choosing the one with the higher score as the model's selection. It has been demonstrated that comparing token probabilities is a more reliable measure of a generative language model's performance compared to free generation [62], and such templates have been successfully used to evaluate bias in LLMs [28].

### 3.5 Two Angles of Model Bias

The VISOGENDER setup has the flexibility to measure model bias in two ways:

**Resolution task** *The resolution task considers a single image with perceived gender label and matches it to multiple candidate captions containing different gender pronouns.* For example, we start with an image containing a doctor perceived as feminine, and specify the set of candidate captions as "the doctor and her/his patient". We define **resolution accuracy**, $RA$, as the percentage of correctly resolved pronouns. This can be calculated over all occupations, across main occupation categories, or per occupation. For a given occupation $o \in O$ and a perceived gender presentation $g$ (either masculine $m$ or feminine $f$), we have:

$$RA_g(o) = \frac{\text{Number of correctly resolved pronouns of gender } g \text{ in occupation } o}{\text{Total number of pronouns of gender } g \text{ in occupation } o}$$

An unbiased outcome is one where the model resolves both perceived gender pronouns equally, i.e., $RA_m(o) = RA_f(o)$, $\forall o \in O$. We now define **resolution bias** as the gender resolution accuracy gap

$$\Delta(o) = RA_m(o) - RA_f(o), \tag{1}$$

where a positive value of $\Delta$ shows a model more accurately resolves masculine-presenting subjects, and vice versa. Our definition of resolution bias measures a form of capability fairness, i.e., whether a system performs equally well across subgroups [8]. This task is applicable to both types of VLMs.

**Retrieval task** *The retrieval task considers a single gender-neutral caption for a given occupation and matches it to multiple images containing subjects with different perceived gender presentations* from the same occupation. For example, we start with the caption "the doctor and their patient" and define the set of candidate images as containing 50% images of doctors who are perceived as masculine and 50% who are perceived as feminine. Given there is no groundtruth for a "correct" ranking of images for a gender-neutral caption, we cannot define a **retrieval accuracy** metric. For defining **retrieval bias**, we use 3 commonly used metrics—*Bias@K* [63], *Skew@K* [64, 11] and *NDKL* [65, 64]. Bias@K measures the overrepresentation of men in the top K retrieval results. Skew@K measures the difference between the desired proportion of image attributes and the observed one, and MaxSkew@K is the maximum Skew among all attributes, or the "largest unfair advantage" [64] belonging to images of any perceived gender presentation. NDKL is a ranking measure that measures the distance from a fair distribution. For further definitions and discussions of these, please refer to the Supplementary Materials. Our definition of retrieval bias measures a form of representational fairness, i.e., with a gender-balanced set of images and a gender-neutral caption, whether each perceived gender group have equal chances of being retrieved. The retrieval task is only applicable to VLEs.

## 4 Results

For the resolution task, we evaluate six VLEs—CLIP [1], OpenCLIP [22] (trained on LAION 2B and 400M [2]), SLIP [23], DeCLIP [24], FILIP [25] (last 3 trained on YFCC-15M [66]); and two state-of-

Table 2: **Resolution bias.** We present resolution accuracy averaged for masculine and feminine gender presentations, as well as the resolution accuracy gap $\Delta$, as defined in eq. (1). "Same perceived presentation of Gender (P.P.Gender)" and "Different P.P.Gender" are images with two people from the same or different P.P.Gender, respectively. A positive gap $\Delta$ denotes better resolution accuracy for masculine-presenting subjects. We also present reported zero-shot classification accuracy on ImageNet [67].

| Model | Overall | Single Person | | Two people | | | | | | ZS Imagenet |
| | | | | Overall | | Same P.P.Gender | | Different P.P.Gender | | |
| | | $RA_{avg}$ | $\Delta$ | $RA_{avg}$ | $\Delta$ | $RA_{avg}$ | $\Delta$ | $RA_{avg}$ | $\Delta$ | |
|---|---|---|---|---|---|---|---|---|---|---|
| CLIP [1] | 0.75 | 0.92 | -0.14 | 0.57 | -0.27 | 0.79 | -0.18 | 0.36 | -0.35 | 63.2 |
| OpenCLIP$_{2B}$ [22] | 0.78 | 0.96 | -0.07 | 0.60 | -0.37 | 0.77 | -0.42 | 0.44 | -0.32 | 66.2 |
| OpenCLIP$_{400M}$ [22] | 0.74 | 0.84 | -0.27 | 0.64 | -0.29 | 0.80 | -0.26 | 0.46 | -0.33 | 62.9 |
| SLIP [23] | 0.60 | 0.77 | 0.14 | 0.43 | 0.14 | 0.51 | 0.12 | 0.34 | 0.17 | 34.3 |
| DeCLIP [24] | 0.70 | 0.87 | 0.06 | 0.52 | -0.17 | 0.74 | -0.14 | 0.29 | -0.19 | 43.2 |
| FILIP [25] | 0.45 | 0.41 | 0.06 | 0.49 | 0.36 | 0.49 | 0.36 | 0.50 | 0.37 | 39.5 |
| BLIP-2 [26] | 0.84 | 0.92 | -0.09 | 0.76 | 0.07 | 0.93 | 0.06 | 0.60 | 0.09 | — |
| GIT [27] | 0.84 | 0.96 | -0.07 | 0.72 | -0.27 | 0.97 | -0.07 | 0.48 | -0.47 | — |

the-art captioning models—BLIP-2 [26] and GIT [27]. For two candidate models (CLIP and BLIP-2, which are among the most downloaded models in the respective model family on Huggingface), we go into more detail by investigating their resolution capabilities and resolution biases, which are also compared to U.S. Labor Force Statistics (Sec. 4.1). We ablate the VISOGENDER setup by changing the order of templates and including a neutral caption. For the retrieval task, we benchmark the same six VLEs on the resolution task. Captioning models are not compatible with the retrieval task. We also compare retrieval bias metrics with U.S. Labor Force Statistics. For all VLEs, we use ViT-B/32 encoders, and for GIT we use the GIT-Large model. We show more detailed analysis for retrieval bias for CLIP and BLIP-2. We present ablation studies and error bars robustness analysis for both tasks in the Supplementary Materials.

## 4.1 Resolution Task

We present results for the resolution task in Tab. 2, disaggregated by different levels of difficulty. We report the mean resolution accuracy $RA_{avg}$ for each difficulty level, together with the resolution bias or accuracy gap $\Delta$. We highlight the difference between model capabilities and model bias—here we evaluate both, where the latter is the gap between model capabilities for perceived genders.

**Evaluating resolution capabilities** As expected, the resolution accuracy is highest when there is one person in the image, and lowest when there are two people of different perceived gender in the image. The accuracies for the latter are consistently worse than random chance, pointing at the models' inability to reason about scenes with multiple people and attributes associated with each of them. This confirms the findings of prior works that conclude that VLMs are not capable of complex visio-linguistic [18] or spatial [68] reasoning. Captioning models are better than, or on par with, VLEs for all levels of difficulty. In Fig. 2, we see that BLIP-2 outperforms CLIP on all perceived presentation of gender splits in the dataset. From Tab. 2 we also see that models with better zero-shot classification accuracy on ImageNet [67] tend to have a better overall resolution accuracy.

**Evaluating resolution bias** From Tab. 2, we see that models tend to exhibit a larger resolution accuracy gap with more difficult subtasks, such as two people with different genders, where there is higher variation and almost random predictions across models. In Fig. 2, we compare the resolution bias, or accuracy gap, for CLIP and BLIP-2. We see that (i) CLIP shows a larger accuracy gap, and (ii) CLIP is more biased towards correctly resolving pronouns for feminine-presenting subjects, whereas BLIP-2 correctly resolves pronouns for masculine-presenting subjects more often. For further analysis and per-occupation results, see the Supplementary Materials.

To interpret the results in a real-world context, we compare U.S. Labor Force Statistics on on proportions of different genders in occupations with resolution bias in Fig. 3. These statistics only account for binary gender, and we have adjusted our perceived masculine and feminine gender presentation counts to be for "men" and "women" respectively. We measure the correlation in the absolute values (with Pearson's R) and correlation in ranked values (with Kendall-Tau), i.e., testing for the monotonicity of relationship between model bias and societal occupation skew [28]. While we see no pattern for the bias of CLIP, the accuracy resolution gap of BLIP-2 correlates with the U.S.

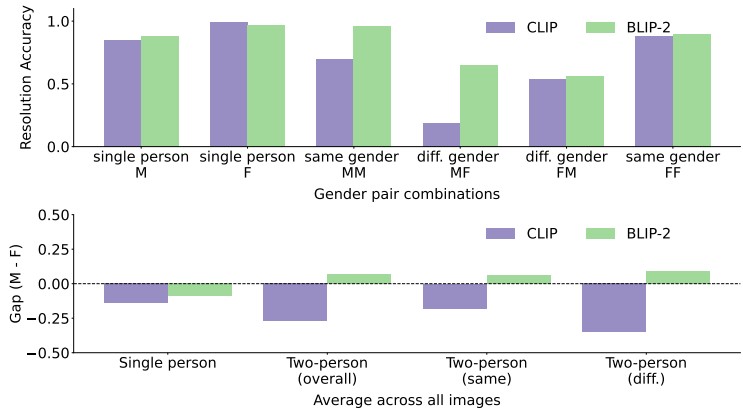

Figure 2: **Resolution accuracy (top) and resolution bias (bottom).** Top: We compare the resolution accuracy for different perceived presentations of gender pair combinations. The lowest RA scores occur in cases with two people with different perceived gender presentations. Bottom: We present the resolution bias (Gap) for either a single person or two people. A bigger Gap score indicates a bias towards one perceived gender presentation. A positive Gap score shows a bias towards masculine-presenting subjects

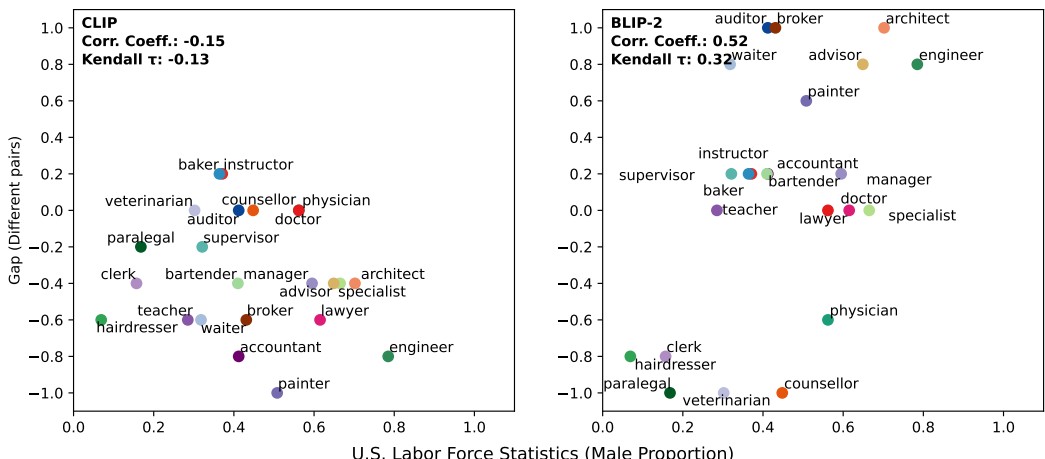

Figure 3: **Resolution bias relative to U.S. Labor Force Statistics.** We compare the gender accuracy Gap per occupation to U.S. Labor Force statistics data. While there are no patterns for CLIP, BLIP-2 shows a bias similar to the real-world data.

proportions—for occupations with fewer women, such as "engineer", the model correctly resolves men more often than women, and vice versa.

## 4.2 Retrieval Bias

We evaluate VLMs on retrieval bias in Tab. 3. We see that *all* models have positive Bias@5 and Bias@10 values, which suggests that images of masculine-presenting subjects in professional settings are retrieved more often than images of feminine-presenting subjects, despite the candidate images always being gender-balanced.

## 5    Discussion

### 5.1    Key Findings

**Models struggle to resolve pronouns in the presence of both perceived presentations of genders**
We found that all VLEs show close to random performance on gender resolution when there are people

Table 3: **Retrieval bias.** We present mean and standard deviation across all occupations. Positive Bias@K shows more images of men were retrieved.

| Model | Bias@5 | | Bias@10 | | MaxSkew@5 | | MaxSkew@10 | | NDKL | |
|---|---|---|---|---|---|---|---|---|---|---|
| | Mean | $\sigma$ | Mean | $\sigma$ | Mean | $\sigma$ | Mean | $\sigma$ | Mean | $\sigma$ |
| CLIP [1] | 0.11 | 0.38 | 0.16 | 0.22 | 0.27 | 0.15 | 0.18 | 0.13 | 0.19 | 0.07 |
| OpenCLIP$_{2B}$ [22] | 0.10 | 0.44 | 0.08 | 0.23 | 0.29 | 0.17 | 0.18 | 0.11 | 0.18 | 0.07 |
| OpenCLIP$_{400M}$ [22] | 0.17 | 0.47 | 0.11 | 0.22 | 0.33 | 0.18 | 0.16 | 0.13 | 0.19 | 0.07 |
| SLIP [23] | 0.06 | 0.52 | 0.00 | 0.24 | 0.32 | 0.21 | 0.17 | 0.12 | 0.19 | 0.09 |
| DeCLIP [24] | 0.11 | 0.40 | 0.15 | 0.26 | 0.28 | 0.16 | 0.20 | 0.14 | 0.17 | 0.07 |
| FILIP [25] | 0.01 | 0.43 | 0.03 | 0.26 | 0.29 | 0.16 | 0.17 | 0.13 | 0.18 | 0.07 |

of different perceived gender presentation in the scene. This hints at insufficient visuo-linguistic capabilities for handling complex scenes in current VLMs.

**Captioning models have a higher accuracy and smaller accuracy gap between genders** We find that captioning models outperform VLEs on all subtasks. We attribute this to the way resolution is done in captioning models—the pronoun of the subject is extracted using the start of the template and next token prediction. Meanwhile, VLEs need to rely on a global `cls` text feature, which seems to not capture the nuanced difference between entities in the sentence.

**Resolution and retrieval bias are not in the same direction** Across models, there is not a consistent pattern of bias direction—VLEs are more accurate at resolving "her" pronouns, while BLIP models are more accurate for "his" pronouns. In contrast, we find that all VLEs are predominately biased towards retrieving images of masculine-presenting subjects. This highlights a risk of representative harm in deploying VLEs in image search systems.

## 5.2 Ethical Considerations

**The harms of performing poorly and exceptionally well on VISOGENDER** In developing this benchmark, we recognise that performing at either end of the spectrum can have harmful side effects. Performing poorly on VISOGENDER can lead to increased gender bias through the use of VLMs when considering the representation of stereotypically feminine binary gender, which is already heavily discriminated in many historically male-dominated industries. For example, if an automatic captioning VLM in a downstream application repeatedly misgenders doctors as "he", this weakens the representation of "women" as doctors too. The risk of erasure and misgendering via pronouns in caption assignment also harms the LGBTQIA+ community. An evaluative tool such as VISOGENDER is designed to flag a model's bias prior to deployment. However, we also recognise that in order to do well on VISOGENDER, a model must perform well not only at scene disambiguation (based on the image subjects' occupational roles) but also have capabilities for recognising (binary) gender. If the VISOGENDER dataset is misused, and does not comply with the terms of use prohibiting its use for training (see Supplementary Materials), there is a potential for increased gender recognition capabilities that can contribute to the development of automatic gender recognition technology. We recognise that the development of this technology denies people—especially transgender, non-binary, and gender-diverse individuals—their dignity, respect, and sometimes safety to exist in public and private spaces. We absolutely condemn any use of VLMs for automatic gender recognition in surveillance use cases.

**Dataset collection with regards to privacy and consent** All images in the VISOGENDER benchmark dataset are collected and used within the scope of their Creative Commons and royalty-free associated licences. However, we respect that these images depict real-life people, that may be misgendered in the development of this benchmark. Importantly, we do not make any claims to correctly assign a person's gender identification and we have included mechanisms for all data subjects to amend the labels and/or request removal of their images.

## 5.3 Limitations

**Subjective assessment when creating datasets** The authors note this work is necessarily influenced by their respective identities. In a random order with psuedo-anonymised identifiers, we present our positionality: A1 is a White Bulgarian cisgendered man; A2 is a White British cisgendered woman;

A3 is a Black British cisgendered woman; A4 is a White South African cisgendered woman; A5 is a Chinese cisgendered man; and A6 is a White Brazilian cisgendered woman. Four authors are pursuing postgraduate degrees and two are pursuing undergraduate degrees, all at the University of Oxford. We draw on our experience in measuring and mitigating social biases in VLMs. Further, when designing, conducting and writing up this research, we consulted closely with domain experts in Gender Studies and members of the LGBTQIA+ community. We acknowledge that visual markers of gender presentation may not reflect a subject's self-identified gender as gender presentation does not necessarily align or reflect in a binary manner with one's sex, pronouns or identity. As such, we acknowledge that gender and gender identity is fluid and exists on a spectrum that is generally misrepresented by binary distinctions. However, irrespective of the societal model of gender, bias exists and leads to representational and allocational harms [8]. This work is not able to address the diversity of gender and other intersectional characteristics that come into the societal image of a typical person doing a certain job as well. However, we attempt at creating a benchmark that could help detect a single level of bias, which is the difference between typical Western presentation of someone perceived as masculine- or feminine-presenting in an occupational role. We advocate that this work is extended to include more genders and avoid erasure of non-binary individuals represented across occupations [69]. The codebase is designed to be flexible to include neopronouns in the future, in order to ensure these systems do not perform poorly when faced with data related to underrepresented communities [70].

**Stacking biases from the internet** We source images from a variety of search platforms (such as Google Image Search) and image hosting sites. While we balance included occupations across perceived presentation of gender search terms, those that we leave out are not "missing at random" due to biases that already exist in images on the internet. We could not find enough images for some occupations, e.g., there were not 5 images of a female-presenting plumber and a male-presenting client that met our criteria for data accessibility and format.

**Dataset representation** This dataset is only intended for evaluation purposes and, as such, requires fewer images than if it were used for training. However, the dataset is still relatively small, and excludes some "non-random" occupations omitted due to existing biases in images on the internet. This exclusion can, in turn, introduce bias into the gender and role depictions. It was beyond our means to partner with a StockImage provider, such as Getty Images [71], but this could be an avenue in future work to expand dataset size and include self-identified pronouns in order to counteract some of the aforementioned image availability biases. Future work could also augment the dataset with synthetic data from generative VLMs [12, 72].

## 6   Conclusion

We introduced VISOGENDER, a novel dataset for benchmarking social biases in VLMs for both pronoun resolution and retrieval settings. On some parts of the benchmark, we demonstrated that current state-of-the-art models perform no better than random chance, and that they do not perform equally well for resolving for both masculine and feminine gender presentations, nor give equal retrieval likelihood to images of masculine- or feminine-presenting professionals. There is significant headroom for improvement both in the reasoning abilities of VLMs, and in the gender gap of their abilities, when it comes to complex scenes with multiple humans. We hope this work encourages the benchmarking of future VLMs, so the risk of downstream harms and negative biases can be measured, compared and mitigated.

## Acknowledgments and Disclosure of Funding

The authors would like to thank the following people for their feedback and insight during the development of VISOGENDER: Max Bain, Hugo Berg, Seb Wilkes, Juliana Mota, Daniel Kochin, Carolyn Dickson, Avishkar Bhoopchand and Rosemary Duffy. This work has been supported by the Oxford Artificial Intelligence student society, the EPSRC Centre for Doctoral Training in Autonomous Intelligent Machines & Systems [EP/S024050/1] (A.S.); the Economic and Social Research Council Grant for Digital Social Science [ES/P000649/1] (H.R.K.); and the Clarendon Fund in partnership with the St Cross College Scholarship, Oxford (F.G.A). For computing resources, the authors are grateful for support from Jonathan Caton and the Google Cloud and the CURe Programme under Google Brain Research.

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

# Appendix

## Contents

# A Retrieval Bias Metrics

Here we define the retrieval bias metrics reported in the paper – Bias@K, MaxSkew@K, and NDKL.

**Bias@K** [63] measures the proportions of individuals with perceived masculine gender presentations and individuals with perceived feminine gender presentations images in the retrievals of a search result with a given text query. For an image $I$, we define a function $g(I) \in \{0, 1\}$ that denotes the perceived gender label of the image:

$$g(I) = \begin{cases} 1 & \text{if the gender of the subject in } I \text{ is perceived as masculine} \\ -1 & \text{if the gender of the subject in } I \text{ is perceived as feminine.} \end{cases}$$

Given a set of $K$ retrieved images $\mathcal{R}_K(q)$ for a query $q$ of a certain occupation, we define the gender bias metric as:

$$\text{Bias@K}(q) = \frac{1}{K} \sum_{I \in \mathcal{R}_K(q)} g(I),$$

and let Bias@K be the average of Bias@K$(q)$ over all occupations $q \in Q$. Bias@K is positive if the retrieval model skews toward professionals perceived with masculine gender presentations in retrieved images, and negative if retrieved for those with perceived feminine gender presentations.

Note that Bias@K approximates a form of representational fairness, and is only applicable to CLIP-like models.

**Skew@K** [64, 11] measures the difference between the desired proportion of image attributes in $\mathcal{R}_k(q)$ for the query $q$ and the actual proportion. Let the desired proportion of images containing a professional of gender $A$ in the set of retrieved images be $p_{d,q,A} \in [0, 1]$ and the actual proportion be $p_{\mathcal{R}(q),q,A} \in [0, 1]$. The resulting Skew@K of $\mathcal{R}(q)$ for a gender $A \in \mathcal{A}$ is:

$$\text{Skew}_A@K(q) = \ln \frac{p_{\mathcal{R}_K(q),q,A}}{p_{d,q,A}},$$

where $p_{d,q,A}$ is the proportion of gender $A$ of occupation $q$ in our dataset, which always equals $0.5$ (gender balanced).

A disadvantage of Skew@K is that it only measures bias with respect to a single gender at a time and must be aggregated to give a holistic view of the bias over all attributes. Following [11], we take the maximum value of Skew@K among all attribute labels $A$ of the retrieved images for the text query $q$:

$$\text{MaxSkew@K}(q) = \max_{A \in \mathcal{A}} \text{Skew}_A@K(\mathcal{R}(q)),$$

which gives us the "largest unfair advantage" [64] belonging to images within a given perceived gender presentation. Here, a MaxSkew@K of 0 for the attribute gender and a given text query $q$ implies that masculine- and feminine-presenting subjects are equally represented in the retrieved set of $K$ images $\mathcal{R}_K(q)$. MaxSkew@K is the average of MaxSkew@K$(q)$ over all occupations $q \in Q$.

**NDKL** [65, 64] (normalised discounted cumulative KL-divergence) measures the distance of the retrieval model from a fair distribution, in a weighted average over $K$. Let $D_{\mathcal{R}_K(q)}$ and $D$ denote the binary distribution of occupational gender over the top $K$ retrieved images and the desired distribution, respectively. Then the NDKL for occupation $q$ is defined as:

$$\text{NDKL}(q) = \frac{1}{Z} \sum_{i=1}^{K} \frac{1}{\log_2(i+1)} D_{\text{KL}}(D_{\mathcal{R}_K(q)} \| D),$$

where $Z$ is a normalising factor $Z = \sum_{i=1}^{K} \frac{1}{\log_2(i+1)}$.

# B   Fine-Grained Resolution Bias Results

In this section, we present fine-grained resolution bias results by sector (Fig. 4), by specialisation (Fig. 5) and by occupation (Fig. 6).

**Bias per sector**   Fig. 4 shows that CLIP is better at resolving perceived feminine gender presentations for most sectors, except for the *medical* sector, where CLIP performs better for perceived masculine gender presentations in some specialisations. BLIP-2, on the other hand, is better at resolving perceived masculine gender presentations in *office* environments.

**Bias per specialisation**   Fig. 5 shows that the specialisations within a sector show similar trends for CLIP, but we observe inter-sector variations for BLIP-2. For example, within the *service* sector, *food* and *household* services are biased towards perceived masculine gender presentations, and *fashion* and *animal* services are biased towards perceived feminine gender presentations. Similarly, for *office* jobs, only *legal* are biased towards feminine gender presentations, and *financial*, *structure*, and *general office* jobs are biased towards perceived masculine gender presentations.

**Bias per occupation**   Fig. 6 shows several occupations with zero accuracy resolution gap – *auditor* for CLIP, and *doctor* and *lawyer* for BLIP-2. For most occupations, CLIP is biased toward correctly resolving perceived feminine gender presentations, and the largest gaps we observe are for *accountant*, *architect*, *engineer*, and *painter*. BLIP-2 is most biased towards correctly resolving perceived masculine gender presentations in the *auditor*, *broker* and *waiter* occupations, and perceived masculine gender presentations in the *counselor*, *paralegal* and *veterinarian* occupations.

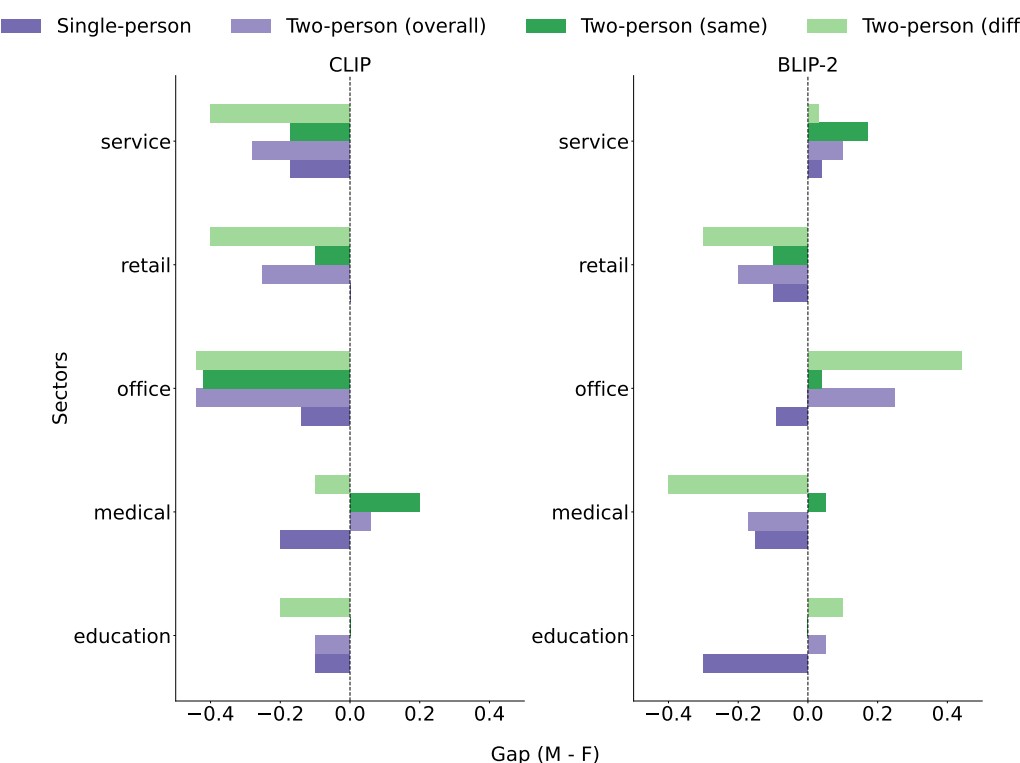

Figure 4: **Resolution accuracy.** Resolution bias (Gap) per Sector

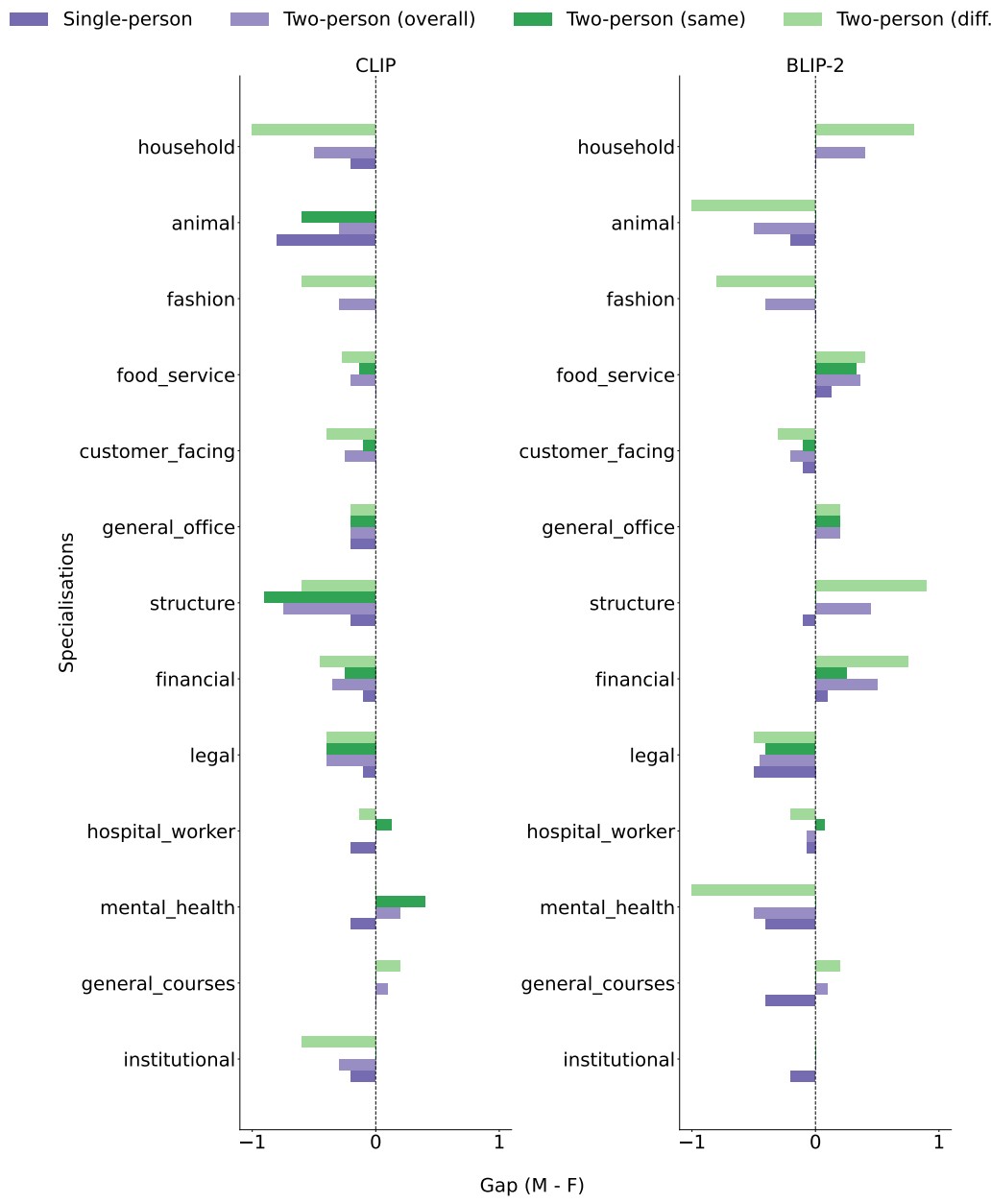

Figure 5: **Resolution accuracy.** Resolution bias (Gap) per Specialisation

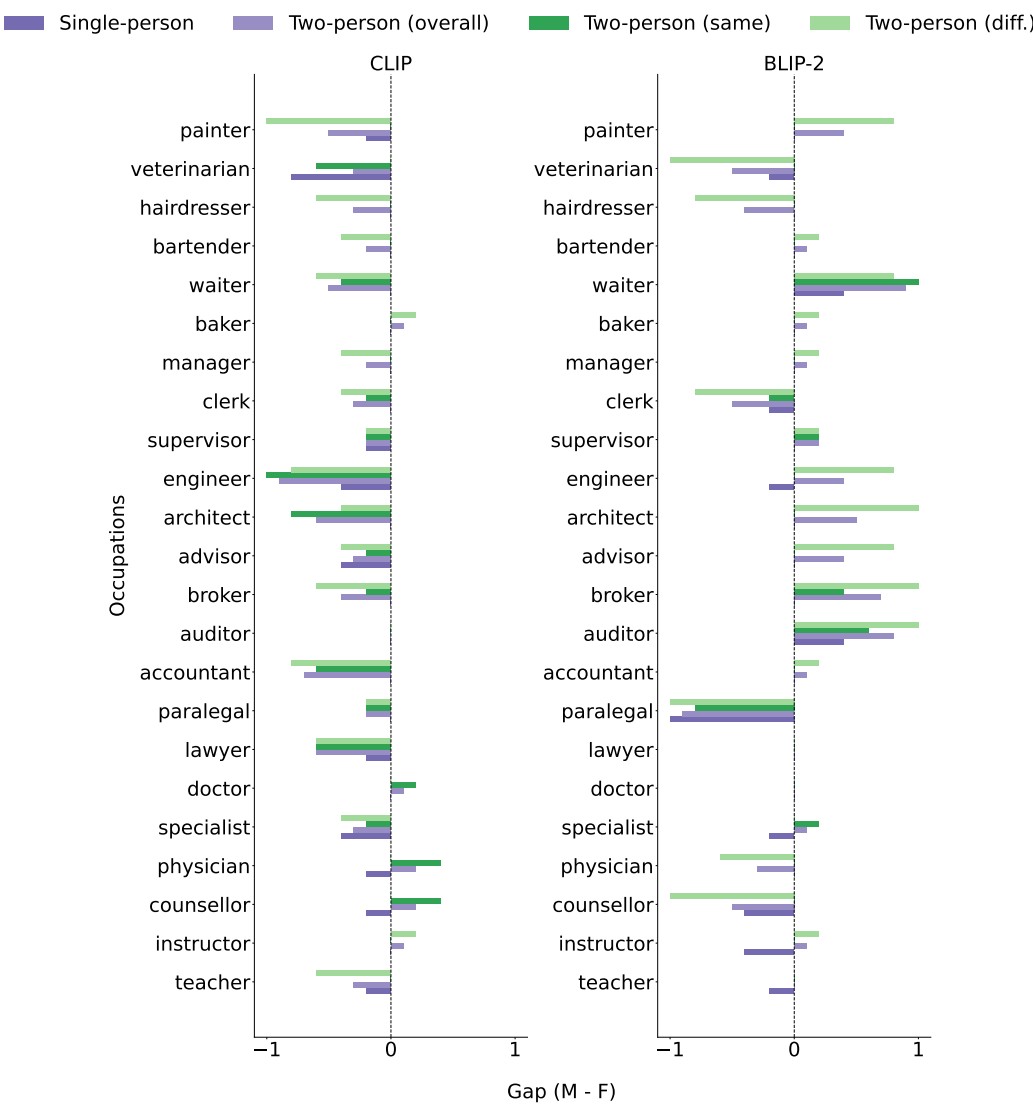

Figure 6: **Resolution accuracy.** Resolution bias (Gap) per Occupation

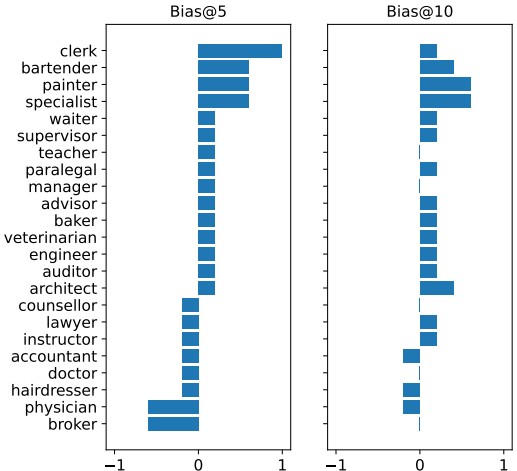

Figure 7: **Retrieval bias by occupation.** We show the bias per occupation of CLIP. Occupations are ranked by the Bias@5 values. Positive Bias@K shows more images of masculine-presenting subjects were retrieved for the given occupation.

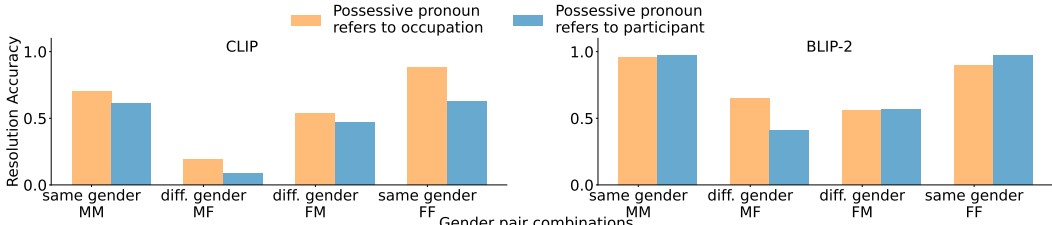

Figure 8: **Resolution accuracy for template flipping.** The resolution accuracy is calculated for all gender pair combinations. The subject of the template refers to either the occupation or the participant.

## C   Fine-grained Retrieval Bias Results

Here we present retrieval bias results per occupation. In Fig. 7, we see that most occupations (15 out of 23) are skewed towards masculine-presenting subjects in both Bias@5 and Bias@10.

## D   Ablations

### D.1   Ablations on Resolution Bias Results

**Template flipping** We change the subject of the prompt sentences for images with two people by reordering the *{participant}* and *{occupation}*, e.g., "The doctor and his patient" becomes "The patient and her doctor". We compare both templates in Fig. 8. While we observe similar trends in the two settings, the resolution accuracy of CLIP is worse when the pronoun refers to the participant.
**Neutral pronoun resolution** Here we attempt to move away from binary gender classification and introduce a third, neutral pronoun – "their", which is always grammatically correct. In Tab. 4 we see

Table 4: **Neutral pronoun resolution.** We measure resolution rates into neutral pronouns. We also show $\Delta_N$, which is the difference between the ratio of images with both masculine and feminine perceived presentations of gender (P.P.Gender) that were resolved with a neutral pronoun.

| Model | Overall | Single Person | | Two people | | | | | |
|---|---|---|---|---|---|---|---|---|---|
| | | | | Overall | | Same P.P.Gender | | Different P.P.Gender | |
| | | $R_{neutral}$ | $\Delta_N$ | $R_{neutral}$ | $\Delta_N$ | $R_{neutral}$ | $\Delta_N$ | $R_{neutral}$ | $\Delta_N$ |
| CLIP [1] | 0.17 | 0.07 | 0.12 | 0.26 | 0.09 | 0.20 | 0.17 | 0.31 | 0.02 |
| BLIP-2 [26] | 0.01 | 0.01 | -0.02 | 0.00 | 0.00 | 0.00 | 0.00 | 0.00 | 0.01 |

Table 5: **Retrieval bias with template flipping.** We present mean and standard deviation across all occupations, when the pronoun refers to the participant in the occupation-participant pair. Positive Bias@K shows more images of masculine-presenting subjects were retrieved.

| Model | Bias@5 Mean | $\sigma$ | Bias@10 Mean | $\sigma$ | MaxSkew@5 Mean | $\sigma$ | MaxSkew@10 Mean | $\sigma$ | NDKL Mean | $\sigma$ |
|---|---|---|---|---|---|---|---|---|---|---|
| CLIP [1] | 0.13 | 0.41 | 0.10 | 0.23 | 0.29 | 0.16 | 0.17 | 0.14 | 0.19 | 0.08 |
| OpenCLIP$_{2B}$ [22] | 0.04 | 0.36 | 0.06 | 0.24 | 0.26 | 0.13 | 0.16 | 0.13 | 0.16 | 0.05 |
| OpenCLIP$_{400M}$ [22] | 0.23 | 0.40 | 0.10 | 0.25 | 0.30 | 0.18 | 0.16 | 0.16 | 0.18 | 0.08 |
| SLIP [23] | -0.18 | 0.49 | -0.06 | 0.26 | 0.35 | 0.18 | 0.17 | 0.14 | 0.21 | 0.09 |
| DeCLIP [24] | 0.15 | 0.49 | 0.13 | 0.27 | 0.35 | 0.16 | 0.20 | 0.14 | 0.19 | 0.08 |
| FILIP [25] | -0.01 | 0.38 | 0.03 | 0.29 | 0.27 | 0.14 | 0.19 | 0.15 | 0.17 | 0.06 |

that while BLIP-2 almost never chooses the neutral pronoun, it is selected by CLIP in 17% of all images and 31% of images containing two people with different perceived gender presentations. We also see that for the more difficult settings, the neutral pronoun is selected more frequently, with 31% in the "two people, different gender" setting, which corresponds to almost random chance (33%). Finally, we see that images, where the perceived gender presentation is masculine, tend to be resolved as neutral more often.

### D.2 Ablations on Retrieval Bias Results

We evaluate retrieval bias when reversing the order in which **{occupation}** and **{participant}** occur. We present results in Tab. 5. While most bias measures are of similar magnitude, Bias@5 and Bias@10 for SLIP are negative – meaning more images of individuals perceived with feminine gender presentations are retrieved. Meanwhile, as reported in the main paper, using the original template, these bias measures were positive.

## E  Results for Additional VLMs

We repeat the analysis in the main paper on additional VLMs – ALIGN [73], FLAVA [74] and GroupViT [75]. These models, similar to the other VLEs evaluated, are biased towards resolving images of individuals with perceived feminine gender presentations more accurately than masculine-presenting subjects, i.e., have a negative gender accuracy gap (see Tab. 6a). The exception is ALIGN in the two-person case, with a slightly positive in the gender accuracy gap. The performance is

| Model | Overall RA$_{avg}$ | Single Person RA$_{avg}$ | $\Delta$ | Two people Overall RA$_{avg}$ | $\Delta$ | Same P.P.Gender RA$_{avg}$ | $\Delta$ | Diff. P.P.Gender RA$_{avg}$ | $\Delta$ | ZS Imagenet |
|---|---|---|---|---|---|---|---|---|---|---|
| ALIGN [73] | 0.52 | 0.78 | -0.33 | 0.26 | -0.02 | 0.42 | -0.01 | 0.10 | 0.03 | 85.5 |
| FLAVA [74] | 0.73 | 0.91 | -0.06 | 0.54 | -0.29 | 0.75 | -0.22 | 0.34 | -0.35 | – |
| GroupViT [75] | 0.64 | 0.79 | -0.16 | 0.50 | -0.09 | 0.64 | -0.11 | 0.34 | -0.07 | – |

(a) **Resolution bias.** We present resolution accuracy averaged for masculine and feminine perceived presenations of gender (P.P.Gender); resolution accuracy gap $\Delta$ as defined in the main paper and zero-shot classification accuracy on Imagenet [67].

| Model | Bias@5 Mean | $\sigma$ | Bias@10 Mean | $\sigma$ | MaxSkew@5 Mean | $\sigma$ | MaxSkew@10 Mean | $\sigma$ | NDKL Mean | $\sigma$ |
|---|---|---|---|---|---|---|---|---|---|---|
| ALIGN [73] | 0.10 | 0.42 | 0.06 | 0.30 | 0.29 | 0.16 | 0.22 | 0.14 | 0.18 | 0.06 |
| FLAVA [74] | 0.10 | 0.47 | 0.11 | 0.20 | 0.31 | 0.18 | 0.16 | 0.11 | 0.20 | 0.07 |
| GROUPVIT [75] | 0.18 | 0.37 | 0.16 | 0.25 | 0.28 | 0.16 | 0.18 | 0.17 | 0.18 | 0.07 |

(b) **Retrieval bias.** We present mean and standard deviation across all occupations for two-person images.

Table 6: Additional bias results.

Table 7: Pearson's correlation coefficients and Kendall $\tau$ when the resolution accuracy results compared to the U.S. Labor Force Statistics (Male Proportion)

| | | Pearson's *r* | | Kendall $\tau$ | |
|---|---|---|---|---|---|
| Occupation \| Participant | | CLIP | BLIP-2 | CLIP | BLIP-2 |
| Same pair | M \| M | -0.45 | 0.40 | -0.31 | 0.36 |
| | W \| W | -0.23 | 0.12 | -0.24 | 0.05 |
| Diff. pair | M \| W | -0.28 | 0.42 | -0.28 | 0.22 |
| | W \| M | -0.05 | -0.53 | -0.06 | -0.42 |

significantly worse in the two-person case, with worst performance for resolution accuracy coming from the setting of different gender subjects. FLAVA has highest overall resolution accuracy, along with the biggest gender gap scores relative to the other two models. Similar to the other VLEs evaluated, we find that the models have a bias toward retrieving images of masculine-presenting subjects in occupational settings (see Tab. 6b).

# F   Results Compared to U.S. Labor Force Statistics

The U.S. Labor Force Statistics publishes an annual report of population demographics across occupations. We record the male/female proportion in the U.S. Labor Force statistics by matching each VISOGENDER occupation to either a single U.S. occupation category or by taking a mean across any relevant categories. For VISOGENDER jobs that cover multiple U.S. categories, we also report the standard deviation. The statisitcs used are predominantly from 2022, with the exception of *medical - specialist* occupation which is taken from 2020. This mapping and accompanying metadata is available in our GitHub repository: https://github.com/oxai/visogender.

## F.1   Evaluating Resolution Bias Relative to U.S. Skew

In Fig. 9 and Fig. 10, we compare the resolution accuracy of CLIP and BLIP-2 per occupation to the proportions of U.S. men working in that occupation. The correlation coefficients for each occupation-participant pair is summarised in Tab. 7. There is no correlation between the resolution accuracy scores and the proportions of the men in the U.S. Labor Force occupation statistics, with the exception of CLIP same pair (M-M) and BLIP-2 different pair (W-M) showing some correlation, that is not significant.

## F.2   Evaluating Retrieval Bias Relative to U.S. Skew

In Fig. 11, we compare the retrieval bias of CLIP to the U.S. male proportions for each occupation. We find that for CLIP, OpenCLIP-400M, and OpenCLIP-2B, the Bias@10 metric (skew toward men) is slightly positively correlated with the proportion of male in each occupation in the U.S., in both Pearson's $r$ and Kendall's $\tau$. Although this correlation is not significant, it reflects potential bias in the methods or dataset of CLIP and OpenCLIP that warrants future investigation.

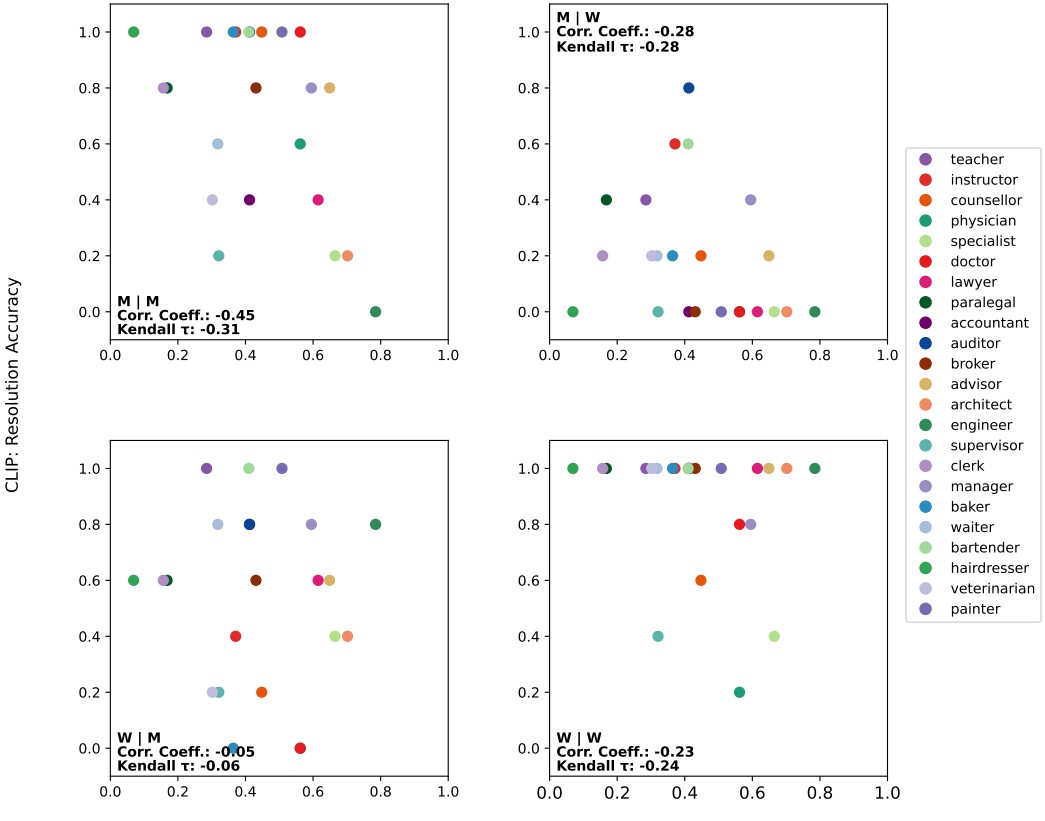

Figure 9: **Resolution accuracy.** CLIP: Mapping of Resolution Accuracy for the same and different gender pairs to the U.S. Labor Force Statistics (Male Proportion)

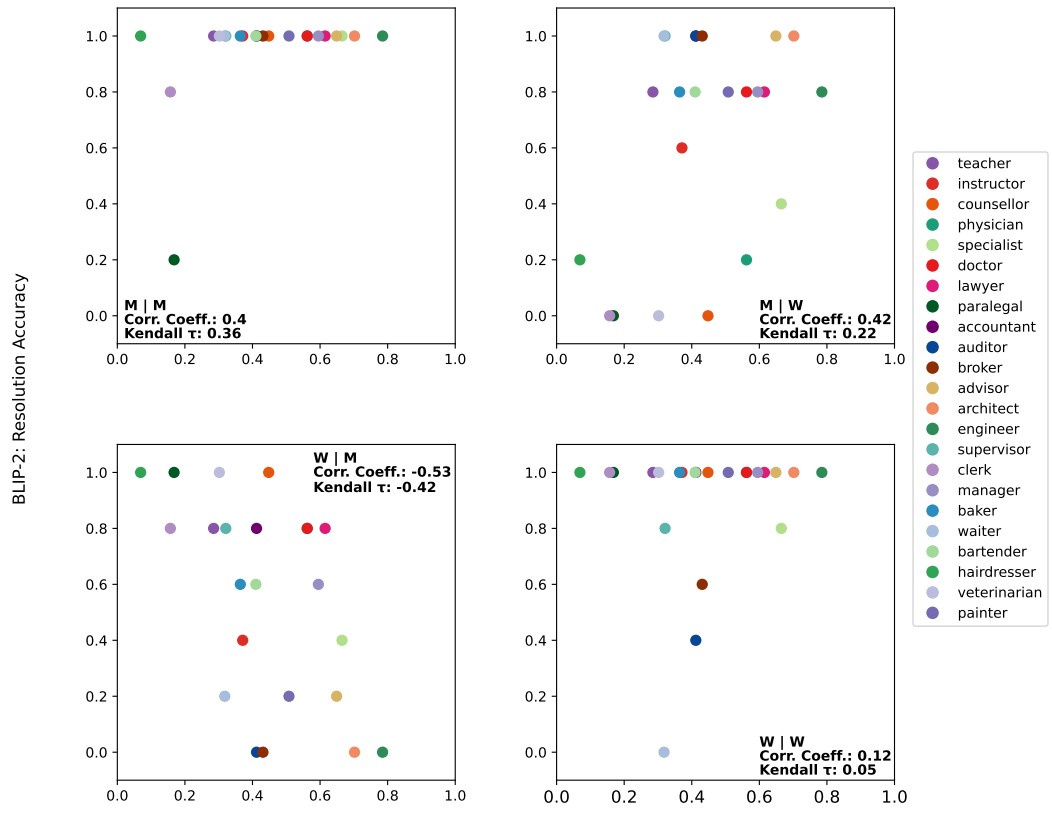

Figure 10: **Resolution accuracy.** BLIP-2: Mapping of Resolution Accuracy for the same and different gender pairs to the U.S. Labor Force Statistics (Male Proportion)

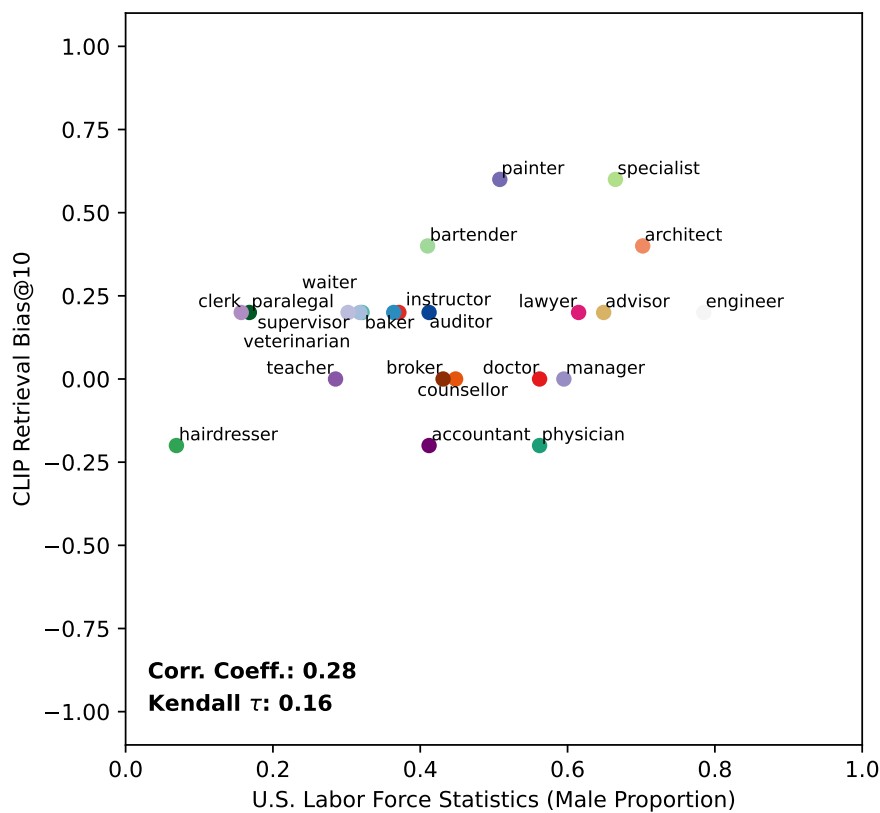

Figure 11: **Retrieval bias.** Comparison of the Bias@10 scores of CLIP to the to the U.S. Labor Force Statistics (Male Proportion).

# G   Error analysis

These analyses are included to ensure the presented results are due to gender bias, and not necessarily from variance in the how many and which images were sampled for inclusion in the dataset.

## G.1   Resolution Bias Error Analysis

In this analysis, we demonstrate that sub-sampling our dataset does not skew the results by much on average. We ran multiple experiments with random sub-samples of the dataset and measured the variance in resulting $\mathrm{RA_{avg}}$ and accuracy gap $\Delta$. In particular, we randomly keep $k$ out of the 5 images per each perceived presentation of gender pair and each occupation, and repeat this trial for 500 times for each $k$ (see Tab. 8). We find that all standard deviations are small ($< 0.02$ in every metric) compared to their mean even if $k = 1$. This means even if we had a dataset with 20% the original size, we would likely have observed the same result (e.g., see Fig. 12). For instance, the difference between models (CLIP, OpenCLIP, etc.) for the same metric is often on a order much larger than 0.02, showing choice of models having a much larger impact on the metrics than the choice or number of images. This shows that despite our dataset being small, it is large enough such that results are significant.

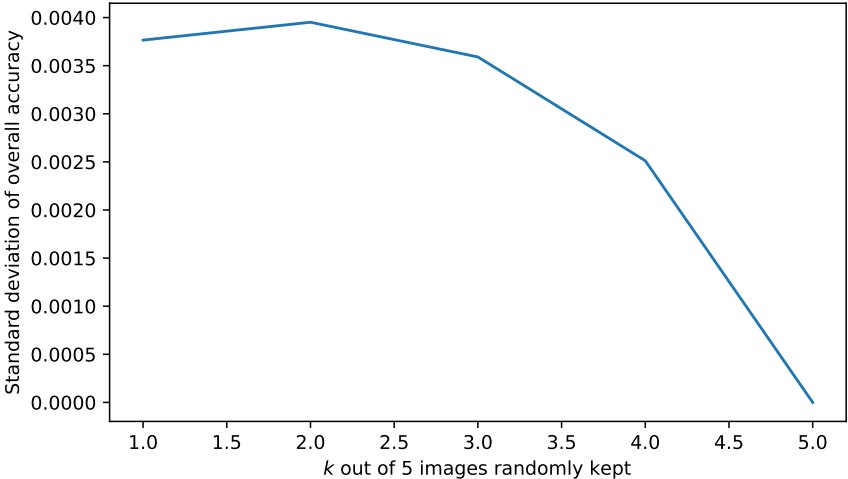

Figure 12: Random analysis showing the variance in overall accuracy (mean $\approx 0.75$) over 500 trials as the number of images kept varies.

## G.2   Retrieval Bias Error Analysis

We compared our existing retrieval bias Bias@10 to those from experiments where: of the 20 images for each occupation, 10 images are randomly chosen and relabeled as "his" and the other 10 as "her" (in other words the top 10 retrieved images are chosen randomly from the 20 images). This is repeated 3000 times, with results presented in Tab. 9. In summary, the average Bias@10 is 0.0003 (which should approach 0), with a standard deviation of 0.0475. Our actual Bias@10 for CLIP is 0.1565,

Table 8: **Resolution bias random sub-samples**. We show mean and standard deviation of each metric, over 500 runs on subsets of our dataset where $k = 1$ image is kept per each group of 5.

| | Overall | Single Person | | Two people Overall | | Same gender | | Different gender | |
| | | $\mathrm{RA_{avg}}$ | $\Delta$ | $\mathrm{RA_{avg}}$ | $\Delta$ | $\mathrm{RA_{avg}}$ | $\Delta$ | $\mathrm{RA_{avg}}$ | $\Delta$ |
|---|---|---|---|---|---|---|---|---|---|
| Mean | 0.7490 | 0.9216 | -0.1392 | 0.5763 | -0.2652 | 0.7871 | -0.1825 | 0.3655 | -0.3479 |
| $\sigma$ | 0.0038 | 0.0051 | 0.0101 | 0.0053 | 0.0113 | 0.0068 | 0.0140 | 0.0082 | 0.0176 |

Table 9: **Retrieval bias random splits.** We present the mean and standard deviation of each metric (in columns), following 3000 runs of random split experiments.

| | Bias@5 | | Bias@10 | | MaxSkew@5 | | MaxSkew@10 | | NDKL | |
| | Mean | $\sigma$ | Mean | $\sigma$ | Mean | $\sigma$ | Mean | $\sigma$ | Mean | $\sigma$ |
|---|---|---|---|---|---|---|---|---|---|---|
| Mean | 0.0014 | 0.3937 | 0.0003 | 0.2271 | 0.2769 | 0.1467 | 0.1504 | 0.1261 | 0.1673 | 0.0609 |
| $\sigma$ | 0.0821 | 0.0563 | 0.0475 | 0.0335 | 0.0307 | 0.0223 | 0.0260 | 0.0164 | 0.0129 | 0.0110 |

which is more than 3 standard deviations away from the expectation. This is repeated for the all metrics (Bias@5, Bias@10, MaxSkew@5, MaxSkew@10, NDKL) and all models (CLIP, OpenCLIP, SLIP, DeCLIP, FILIP), and most of them show the models being more biased than random retrievals (mostly at least 1 standard deviation away). This shows that our models are more biassed when we split by perceived gender presentation, as opposed to randomly splitting.

## H  VISOGENDER Dataset Criteria

Data collection was carried out by the authors of the paper from March to May 2023 on a variety of image databases and search providers, such as Pexels and Google Image Search. We followed a set of guidelines to specify exclusion and inclusion criteria:

1. There is either only one or two people in the image, depending on the task,
2. The image does not contain any children,
3. The images are safe-for-work,
4. The image is a photograph (in colour or black and white) but not e.g., a cartoon,
5. The image does not contain stock photo watermarks,
6. The image can be accessed by a URL,
7. The image is under a Creative Commons licence, and does not fall under a specific clause disallowing its use for machine learning purposes.

## I  VISOGENDER Occupational Taxonomy

The taxonomy, as indicated in Fig. 13, is characterised according to a three tier system: **Sector** (the general field), **Specialisation** (subcategories within the Sector) and **Occupation** (job categories nested within the Specialisation). Tab. 10 outlines the number of images per Sector and Specialisation. category.

Table 10: **Dataset statistics.** Number of images per sector and specialisation.

|  |  | **Single-person** {Occuptation}–{Object} | **Two-person** {Occuptation}–{Participant} |
|---|---|---|---|
| **Sector** | Education | 20 | 40 |
|  | Medical | 40 | 80 |
|  | Office | 90 | 180 |
|  | Retail | 20 | 40 |
|  | Service | 60 | 120 |
| **Specialisation** | General courses | 10 | 20 |
|  | Institutional | 10 | 20 |
|  | Mental heath | 10 | 20 |
|  | Hospital worker | 30 | 60 |
|  | Legal | 20 | 40 |
|  | Financial | 40 | 80 |
|  | General office | 10 | 20 |
|  | Structure | 20 | 40 |
|  | Customer facing | 20 | 40 |
|  | Food service | 30 | 60 |
|  | Fashion | 10 | 20 |
|  | Animal | 10 | 20 |
|  | Household | 10 | 20 |

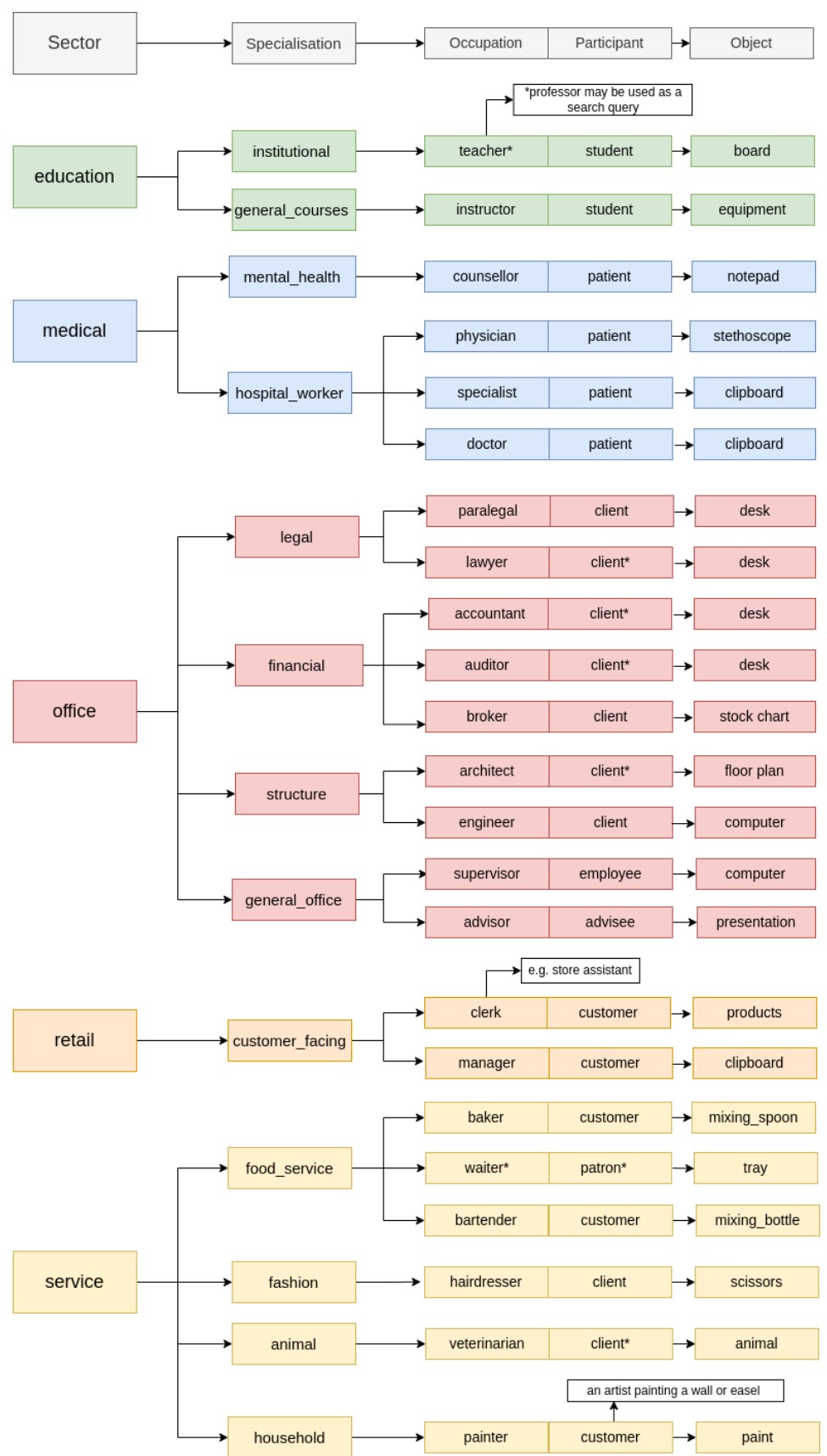

Figure 13: **Taxonomy of the VISOGENDER dataset based on Winogender occupations [19].** All participant roles indicated by a * have been adapted from Winogender.

## J   How To Guide for Benchmarking a VLM

A VLM can be evaluated on our benchmark using the provided code (https://github.com/oxai/visogender). The benchmark scores are saved in JSON files, as depicted below.

To perform well on VISOGENDER, the scores should be optimised as follows:

- Resolution bias:
    - resolution accuracies should be as close to 100% as possible indicating a high capability in performing gender coreference resolution
    - the gender gap score should be as close to zero as possible to ensure the model is not biased towards either gender
- Retrieval bias: all scores should be as close to zero as possible to demonstrate a model is not biased towards either gender.

```
{
  "visogender_blip2_results": {
    "metadata": {"experiment_desc": "CAPTIONING", "model_name": "blipv2"},
    "resolution_bias": {
      "all_images": {"overall_accuracy": 0.84},
      "single_person_images": {"RA_avg": 0.92, "gender_gap": -0.09},
      "two_person_images": {"RA_avg": 0.76, "gender_gap": 0.07},
      "two_person_images_same_gender": {"RA_avg": 0.93, "gender_gap": 0.06},
      "two_person_images_diff_gender": {"RA_avg": 0.60,"gender_gap": 0.08}
    }
  }
}
```

```
{
  "visogender_clip_results": {
    "metadata": {"experiment_desc": "CLIP","model_name": "clip"},
    "resolution_bias": {
      "all_images": {"overall_accuracy": 0.75},
      "single_person_images": {"RA_avg": 0.92, "gender_gap": -0.14},
      "two_person_images": {"RA_avg": 0.57, "gender_gap": -0.27},
      "two_person_images_same_gender": {"RA_avg": 0.79, "gender_gap": -0.18},
      "two_person_images_diff_gender": {"RA_avg": 0.36, "gender_gap": -0.35}
    },
    "retrieval_bias": {
      "bias@5": {"mean": 0.13, "sigma": 0.35},
      "bias@10": {"mean": 0.16, "sigma": 0.22},
      "maxskew@5": {"mean": 0.25, "sigma": 0.15},
      "maxskew@10": {"mean": 0.18, "sigma": 0.13},
      "NDKL": {"mean": 0.18, "sigma": 0.07}
    }
  }
}
```

# K   VISOGENDER Data Clause

## K.1   Terms of Use

This dataset is solely intended for use as a benchmark for evaluating vision-language models under the constraints of the licence. This dataset is strictly not to be used for training under any circumstances.

## K.2   Licence

The VISOGENDER dataset only contains URLs that reference images that (at the time of curation) are under Creative Commons and/or royalty free licences that allow for their use and distribution. No images are stored directly. The VISOGENDER dataset is bound under a CC-BY 4.0 licence and is used as such by the authors. However, it is important to note that individual images in the model may have licences that do not allow commercial use, and users of this dataset will assume liability if they use the dataset beyond the terms of use as indicated by the benchmark. The authors do not take responsibility for any licences that change with time.

The authors confirm that, to the best of their knowledge, they are using all intellectual property in accordance with their licences, and the use of the data as stipulated in this terms of use and the accompanying manuscript and GitHub repository does not violate any rights. The GDPR allows the processing of personal data for research purposes and only includes the URLs so there is no personal data shared.

## K.3   Dataset Maintenance

The URLs are curated manually, and at the time of collection in April - May 2023, they did not point to any images containing harmful or disturbing imagery, nor do any images depict children. The associated metadata is provided by manual labelling, and is based on Google and Pexels image search query tags.

The authors undertake to do the following:

- The authors will proactively investigate the dataset for broken links, with randomised checks of the images themselves to ensure URLs are not redirecting every 6 months
- We have uploaded code and instructions to GitHib for easy command line running of a script which checks for URL integrity and that the images can be utilised by models. This will be run by the authors every 6 months

We also welcome feedback and scrutiny from the community making use of the benchmark. In order to facilitate this process, we put forward the following:

- We have included instructions for running the code to check the dataset,
- Further, we have included a Google Form which can be used to identify broken and/or inappropriate links

## K.4   Reporting and/or Addressing Issues with the Dataset

In the event that there are any issues with the dataset, or any specific links to images, or associated images, please contact the authors by filling out this Google Form and any offending information will be removed immediately. These issues can include, but are not limited to issues with deprecated links, links that have redirected to disturbing, inappropriate content, or you would like images related to yourself, personally removed.

# L   VISOGENDER Datasheet

We present a Datasheet for the VISOGENDER dataset [59] which is available on GitHub: https://github.com/oxai/visogender/. The information in the datasheet is up to date as of June 2023. Any amendments to the datasheet subsequent to this version will be made on GitHub.

### L.1 Motivation

**For what purpose was the dataset created?**    The dataset was created as a benchmark to evaluate vision-language models (VLM). Specifically, it is designed to measure a model's abilities and biases for gender pronoun coreference resolution and image retrieval in an occupational setting.

**Who created the dataset (e.g., which team, research group) and on behalf of which entity (e.g., company, institution, organisation)?**    The authors of the paper, as a part of the Oxford Artificial Intelligence Society Research Labs.

**Who funded the creation of the dataset?**    N/A

**Any other comments?**    None.

### L.2 Composition

**What do the instances that comprise the dataset represent (e.g., documents, photos, people, countries)?**    The dataset comprises images of people in a professional setting. There is at least one person in the image which represents the professional person as referred to by their occupation.

**How many instances are there in total (of each type, if appropriate)?**    Single person: 230 ∥ Two people in the image: 460 ∥ **Total number of images: 690**. See Tab. 1 for more information.

**Does the dataset contain all possible instances or is it a sample (not necessarily random) of instances from a larger set?**    This images were curated from Google image search (Creative Commons Licence) and Pexels images which are royalty free. In most cases, we took the first 5 images per gender-pair and occupation meeting the criteria and guidelines. Our restrictive licensing criteria and resource constraints (i.e., limiting partnerships with StockPhoto providers) meant that the set of total images to chose from was limited. In some cases, there were insufficient images that met the criteria. However, compared to all images on the internet of individuals in professional settings, this is a non-random subsample.

**What data does each instance consist of?**    Each image contains one person, as referred to by their occupation, or, in the case of two people: one person referred to by their occupation and another referred to by a participation noun, relative to the interaction with the professional. In the case of one person in the image, they are accompanied by an object.

**Is there a label or target associated with each instance?**    Yes, there is a perceived presentation of gender label and the labels refer to the professional person and their participant (if applicable). There is an object label as well, in the case of a single person being present. There are also labels related to the occupation, based on the taxonomy.

**Is any information missing from individual instances?**    No.

**Are relationships between individual instances made explicit (e.g., users' movie ratings, social network links)?**    N/A

**Are there recommended data splits (e.g., training, development/validation, testing)?**    This is purely an evaluation set. It is not intended for training purposes.

**Are there any errors, sources of noise, or redundancies in the dataset?**    At the time of initial release, there are no errors, redundancies or sources of noises to the best of the authors' knowledge, based on internal review.

**Is the dataset self-contained, or does it link to or otherwise rely on external resources (e.g., websites, tweets, other datasets)?**    The dataset contains URLs that reference images that (at the time of curation) are under Creative Commons and/or royalty free licences that allow for their use and distribution.

**Does the dataset contain data that might be considered confidential (e.g., data that is protected by legal privilege or by doctor–patient confidentiality, data that includes the content of individuals' non-public communications)?** The data collected (URLs) does contain identifiable information. The metadata doesn't contain any personal identifiable information such as names, emails etc.. However, the images are only pseudonymised as the faces are recognisable. However, they are publicly available under Creative Commons and royaty-free licenses.

**Does the dataset contain data that, if viewed directly, might be offensive, insulting, threatening, or might otherwise cause anxiety?** The URLs are curated manually, and at the time of collection, they did not point to any images containing harmful or disturbing imagery, nor any images containing children. Any URL endpoints that change to become problematic or are determined to infringe on privacy will be removed immediately.

**Does the dataset identify any subpopulations (e.g., by age, gender)?** The images are of people with identifiable features that can infer such information. However, no identifiable or personal information is shared with the dataset. However, we do infer a perceived gender presentation that is necessarily influenced by our internal biases and we acknowledge we risk misgendering a person depicted in the image. We have included opt-out mechanisms and the option to report any adjustments to our labels. Please see this Google Form. The authors are notified immediately when an entry is submitted to the form.

**Is it possible to identify individuals (i.e., one or more natural persons), either directly or indirectly (i.e., in combination with other data) from the dataset?** The images depict real people who can be identified but there is no text data linked to their physical image. These images are publicly available under Creative Commons and/or royalty free licences, and our terms of use, and use of VISOGENDER in research is not in violation of these licences. We rely on the informed consent processes to which we do not have access.

**Does the dataset contain data that might be considered sensitive in any way (e.g., data that reveals race or ethnic origins, sexual orientations, religious beliefs, political opinions or union memberships, or locations; financial or health data; biometric or genetic data; forms of government identification, such as social security numbers; criminal history)?** Only race can potentially be inferred from the images. To the best of the authors' knowledge, these public images were not collected alongside this sensitive information.

**Any other comments?** We have included opt-out options and the option to request amendments to the labelling process. This is available in our GitHub repository: `https://github.com/oxai/visogender`.

## L.3 Collection Process

**How was the data associated with each instance acquired?** Search queries such as "a photo of a {gendered adjective} {occupation} and a {gendered adjective} {participant}" or "a photo of a {gendered adjective} {occupation} and a {object}" were used on Google image search (with the Creative Commons licence filter) and Pexels image search.

**What mechanisms or procedures were used to collect the data (e.g., hardware apparatuses or sensors, manual human curation, software programs, software APIs)?** Image search queries, and manual labelling using a shared Google Sheets document.

**If the dataset is a sample from a larger set, what was the sampling strategy (e.g., deterministic, probabilistic with specific sampling probabilities)?** N/A

**Who was involved in the data collection process (e.g., students, crowdworkers, contractors) and how were they compensated (e.g., how much were crowdworkers paid)?** The authors of the paper were responsible for the data collection, and no compensation was provided. No external parties were involved.

**Over what timeframe was the data collected?**    Two months in 2023 (March to May).

**Were any ethical review processes conducted (e.g., by an institutional review board)?**    No, as the images are under open licences, and no human subjects were recruited for the purpose of the study.

**Did you collect the data from the individuals in question directly, or obtain it via third parties or other sources (e.g., websites)?**    Third party websites, with appropriate licences allowing us to use the images.

**Were the individuals in question notified about the data collection?**    No, as we are collating images collated by third parties. However it is assumed, based on the associated licences, that the individuals depicted were made aware that their images were taken for public access.

**Did the individuals in question consent to the collection and use of their data?**    Please see the previous question.

**If consent was obtained, were the consenting individuals provided with a mechanism to revoke their consent in the future or for certain uses?**    Any images will be removed immediately if there are any objections. To facilitate this process, we have a Google Form which can filled out and the authors are notified by email.

**Has an analysis of the potential impact of the dataset and its use on data subjects (e.g., a data protection impact analysis) been conducted?**    No.

**Any other comments?**    The authors confirm that, to the best of their knowledge, they are using all intellectual property in accordance with their licences, and the use of the data does not violate any rights. The authors do not take responsibility for any licences that change with time.

### L.4    Preprocessing/Cleaning/Labelling

**Was any preprocessing/cleaning/labelling of the data done (e.g., discretisation or bucketing, tokenisation, part-of-speech tagging, SIFT feature extraction, removal of instances, processing of missing values)?**    Manual labelling of the data was conducted at the time of the data collection. Images were also sorted into an occupational taxonomy we created.

**Was the "raw" data saved in addition to the preprocessed/cleaned/labelled data (e.g., to support unanticipated future uses)?**    The only data that was collected is present in the open version of VISOGENDER. The data only includes URls and the VISOGENDER metadata, but no images are downloaded.

**Is the software that was used to preprocess/clean/label the data available?**    No specific software was used during data collection. However, there is code to benchmark models over the dataset via the URLs and metadata files.

**Any other comments?**    No.

### L.5    Uses

**Has the dataset been used for any tasks already?**    The dataset is completely novel and, as of October 2023, has only been used in the original VISOGENDER paper.

**Is there a repository that links to any or all papers or systems that use the dataset?**    Yes. Please see our GitHub repository.

**What (other) tasks could the dataset be used for?**    The dataset should only be used as an evaluation dataset for binary gender bias vision-language models. It is not advised to use this dataset for training purposes, as the binary nature of the labels risks erasure of non-binary people represented in these professional roles.

**Is there anything about the composition of the dataset or the way it was collected and prepro- cessed/cleaned/labelled that might impact future uses?** We only collect URLs which link to images hosted on third party sites. There is the potential that these links become deprecated, or image licences change over time. The authors do not take responsibility for any changes to image links, or their associated licences, but will immediately remove any problematic images in the event these are identified.

**Are there tasks for which the dataset should not be used?** This should not be used for the training of models. It is solely intended as an evaluation dataset.

**Any other comments?** None.

## L.6 Distribution

**Will the dataset be distributed to third parties outside of the entity (e.g., company, institution, organisation) on behalf of which the dataset was created?** This dataset is publicly available and it is encouraged that developers of VLMs use it to assess their models' propensities for gender bias.

**How will the dataset be distributed (e.g., tarball on website, API, GitHub)?** GitHub.

**When will the dataset be distributed?** At the time of the paper being published in 2023.

**Will the dataset be distributed under a copyright or other intellectual property (IP) licence, and/or under applicable terms of use (ToU)?** Yes. We have included a Data Clause which includes the licence and terms of use in the Supplementary Materialsand in our GitHub repository under "LICENCE": https://github.com/oxai/visogender/blob/main/LICENCE. These URLs are distributed based on their royalty free/ Creative Commons licences that the images occupy at the time of curation. The dataset is open source, but we request that the dataset is cited in any subsequent work. The citation can be found alongside the data on our GitHub repository: https://github.com/oxai/visogender.

**Have any third parties imposed IP-based or other restrictions on the data associated with the instances?** No, not at the time of data curation (March - May 2023).

**Do any export controls or other regulatory restrictions apply to the dataset or to individual instances?** No.

**Any other comments?** None.

## L.7 Maintenance

**Who will be supporting/hosting/maintaining the dataset?** The authors of the paper and the Oxford Artificial Intelligence Society.

**How can the owner/curator/manager of the dataset be contacted (e.g., email address)?** The first author (Siobhan Mackenzie Hall) can be contacted by email (siobhan.hall@nds.ox.ac.uk), or via a GitHub Issue: https://github.com/oxai/visogender/issues. Alternative, any issues can be logged on the Google Form which is also linked in the README.

**Is there an erratum?** N/A at the time of publishing.

**Will the dataset be updated (e.g., to correct labelling errors, add new instances, delete instances)?** Yes. We encourage anyone that finds fault to contact the authors to amend or remove any problematic URLs. Alternatively, please complete this Google Form to report problematic images and/or labels and the authors will be notified by email immediately. The authors undertake to do the following:

1. The authors will proactively investigate the dataset for broken links, with randomised checks of the images themselves to ensure URLs are not redirecting every 6 months

2. We have uploaded code and instructions to GitHub for easy command line running of a script which checks for URL integrity and that the images can be utilised by models. This will be run by the authors every 6 months

3. Further, we have included a Google Form which can be used to identify broken and/or inappropriate links

**If the dataset relates to people, are there applicable limits on the retention of the data associated with the instances (e.g., were the individuals in question told that their data would be retained for a fixed period of time and then deleted)?**  N/A as the images are under Creative Commoms and/or royalty-free licences that don't prohibit the use of images for machine-learning purposes.

**Will older versions of the dataset continue to be supported/hosted/maintained?**  Yes, with the exception of any URLs that change with time and potentially link to problematic images.

**If others want to extend/augment/build on/contribute to the dataset, is there a mechanism for them to do so?**  Yes. We encourage anyone looking to expand the dataset to contact the authors to discuss this development.

**Any other comments?**  None.

