# OpenReview forum: "VisoGender:  A dataset for benchmarking gender bias in image-text pronoun resolution"
_NeurIPS.cc/2023/Track/Datasets_and_Benchmarks — NeurIPS 2023 Datasets and Benchmarks Poster_

### Official Review · Reviewer_BTLm · 2023-07-20
**Well-written paper but problematic benchmark design and documentation**

**Rating:** 5
**Confidence:** 4

**Strengths:**

- The authors concretely ground the biases of VLMs in real-world harms (e.g., sexist societal perceptions of women, misgendering), thereby motivating their benchmark well.

- Per the authors' claims, VisoGender is the first benchmark to evaluate gender biases in visual-linguistic coreference resolution. Hence, this paper is relevant to the bias evaluation community.

- The authors' conception of bias as occupation-related gender bias is relatively clear [5].

- The authors' finding that resolution and retrieval bias are not in the same direction across models is interesting. Do the authors have any hypotheses for why this was observed?

**Additional Feedback:**

References:

[1] https://ironholds.org/resources/papers/gender_multiplicity.pdf

[2] https://dl.acm.org/doi/10.1145/3274357

[3] https://www.queerinai.com/naiac-briefing

[4] https://aclanthology.org/2021.acl-long.81/

[5] https://aclanthology.org/2020.acl-main.485/

[6] https://aclanthology.org/2021.emnlp-main.150.pdf

**Clarity:**

- The paper is well-organized and clearly written.

**Correctness:**

- The benchmark is fundamentally problematic; models cannot, and should not, predict the pronouns of individuals from scenes. The benchmark collapses the multiplicity of gender [1]; it assumes that pronouns can be inferred from U.S.-normative gender presentation, and further posits that such inference is important to visual-linguistic reasoning. In fact, the authors even incorrectly conflate "groundtruth gender" and pronouns (lines 110-111) [6]. This benchmark is adjacent to automatic gender recognition, which numerous researchers have warned against due to its potential to misgender, out, and surveil LGBTQIA+ people [2, 3].

- Lines 300-302: The solution here is NOT to include non-binary people or neopronouns as additional categories; inferring gender and pronouns is problematic regardless.

- Lines 38-39: How do annotators guess the gender or pronoun of image subjects? This is not just a "complicated societal matter" (line 299); it is unethical.

- The authors use appropriate baseline models and metrics (e.g., accuracy gap, Bias@K) to measure bias.

**Documentation:**

- The authors do an excellent job of documenting where they rely on previous work (e.g., for their list of occupations) and any modifications they make.

- The authors provide a thorough datasheet.

- The authors provide clear background information and usage instructions on their GitHub. However, their repository does not contain a license for their benchmark.

- The authors do not discuss consent or copyright concerns regarding the images in their benchmark. In particular, image subjects may not consent to: 1) being included in the benchmark, and 2) their inferred gender annotations.

**Ethics:**

- This paper requires an ethics review. Please refer to the Correctness and Documentation comments.

**Limitations:**

- While the authors claim that VisoGender can measure biases across multiple downstream tasks, the external validity of the scenes they curate and the natural language captions they construct is questionable (e.g., it is unclear how well bias evaluation on these scenes with only 1-2 people and synthetic captions transfer to real-world images and captions) [4]. The authors should employ a more reflexive lens throughout the paper.

- The authors should reflect more critically on their decision to filter out occupations for which they could not find sufficient image data (lines 158-159). What are the concrete, real-world implications of doing so? Occupations for which there is significant representational bias in existing vision-language retrieval engines are arguably the most important to include in the benchmark.

**Opportunities For Improvement:**

- The authors should make their context and positionality clearer, and how these affect their benchmark design. Does their benchmark center Western professions and appearances thereof (this is likely given that they source data from Google Images and compare their results to U.S. labor force statistics)? Does their benchmark center Western-normative gender presentation? Given social and historical context, what kinds of occupation-related gender bias do the authors expect models to have a priori?

- Combining WinoGender schema with scenes is interesting, but it feels like a rather incremental approach to detecting bias in VLMs; what other caption or scene curation/design choices did the authors consider but ultimately not employ, and why?

- Line 93: Besides balancing gender, what other steps were taken to minimize spurious correlations? What would be examples of spurious correlations in the context of this benchmark?

- Line 135: How is the subject vs. object of images decided? While the subject/object entities are flipped (e.g., "the doctor and her patient" and "the patient and his doctor") in the ablation, why are they not randomly flipped in the benchmark itself?

- Line 282: The authors should comment of if/how it is possible to interpret resolution accuracy gaps when the overall accuracy of resolution is low.

- Discussion: The authors should clearly state what is entailed by a model performing poorly vs. well on their benchmark.

**Relation To Prior Work:**

- The authors claim that, unlike prior work, VisoGender can measure biases across multiple downstream tasks, e.g., captioning, image retrieval.

- The authors clearly trace the origins of their benchmark to the WinoGround and WinoGender benchmarks.

**Summary And Contributions:**

The authors introduce the VisoGender benchmark for occupation-related gender biases in vision-language models (VLMs). The benchmark comprises scenes with WinoGender-like captions, and it evaluates bias by comparing: 1) gender resolution accuracies for men and women, and 2) ratios of male and female professionals retrieved for a gender-neutral search query. Using their benchmark, the authors show that several SOTA vision-language models cannot correctly resolve gender in complex scenes; the authors further find that captioning models are often more accurate and less biased than CLIP-style models.

---

> ### Author Response · Authors · 2023-08-17
> **Response Part 1/2: Strengths, Opportunities for Improvement**
>
> Thank you very much for your thorough review of our paper. We hope that our responses address the seriousness and importance of these concerns. We would like to emphasise that we are absolutely against the use of automatic gender recognition and surveillance technologies, and recognise the severe harms these impose on gender-diverse, LGBTQIA+ and other vulnerable communities. However, with this benchmark, we believe it is necessary to acknowledge the extreme prevalence of (binary) gender bias in the VLMs when it comes to representation of typically feminine binary gender (‘femininity’) which is already heavily discriminated against in many historically men-dominated industries. We have carefully revised the paper to reflect on and discuss your concerns.
>
> If there are any additional suggestions you may have to make our stance clearer, we’d welcome your suggestions.
> Please see our responses to your specific questions below.
>
> **Strengths:**
>
> **S1: The authors' finding that resolution and retrieval bias are not in the same direction across models is interesting…**
>
> **SR 1:** The resolution bias measures the accuracy of predicting an inferred pronoun from  perceived gender presentation of an individual in a given setting, whereas retrieval bias measures the association between an occupation and an image of a person in the occupation. Similarly to the evaluation metrics for VLMs, zero-shot classification accuracy, and ranking score (recall@K), these two correlate in general, but do not necessarily go in the same direction. Further investigations are needed on the way these biases emerge.
>
> **Opportunities for improvement (OI)**
>
> **OI 1: The authors should make their context and positionality clearer, and how these affect their benchmark design**
>
> **OI R1:** As per this suggestion, we have added a positionality statement and acknowledged the limitations of our biases that affect image curation, annotation and research design  (see page 10 L352-360).
>
> This benchmark presents a Western-centric, cisgendered and stereotypical view of binary gender. The types of jobs and their image representations are also stereotypical of Western-centric occupations. We expect the models a priori to favour genders that have been typically over-represented for the corresponding roles, for example the law profession primarily exhibiting over-representation of   individuals perceived as men and domestic or school teaching roles exhibiting over-representation of  individuals perceived as women.
>
> **OI 2: …incremental approach to detecting bias in VLMs; what other caption or scene curation/design choices did the authors consider but ultimately not employ, & why?**
>
> **OI R2:** This work is first of its kind for VLM and so an incremental start is still useful, as we aim for this to provide a baseline for future work. We build directly on impactful work in NLP but introduce novel tasks (retrieval and resolution) and metrics to adapt to the vision-language context.
>
> We considered alternative scene curation options, such as staging some scenes using the authors as subjects, but ultimately decided this would be too artificial and chose existing images “in the wild”. Our design choice was tightly coupled to Winogender [1] from the beginning, and this guided our development.
>
> **OI 3: Line 93: Besides balancing gender, what other steps were taken to minimise spurious correlations?**
>
> **OI R3:** We relied on stereotypical clothing and equipment and objects associated with the occupation. We avoided using the same people depicted in multiple scenes, to increase diversity of the people in terms of age and race, as well as the backgrounds in the images.
>
> **OI 4: How is the subject vs. object of images decided?**
>
> **OI R4:** We rely on the clothing and contextual clues in the image, with the lens of a Western presentation of these scenes. The participant is typically in casual clothing, while the person in an occupational role will have clothing and equipment stereotypical of that role. The benchmark only includes the occupation first template for two reasons:
> (1) there are related occupational stats
> (2) it's better studied in prior work (within Comp. Sci and also in social scientific/gender studies / historical gender discrimination etc).
>
> **OI 5:** Line 282: The authors should comment of if/how it is possible to interpret resolution accuracy gaps when the overall accuracy of resolution is low
>
> **OI R5:** This relative difference still provides us with a better baseline than random chance.
>
> **OI 6: Discussion: The authors should clearly state what is entailed by a model performing poorly vs. well**
>
> **OI Response 6:** This is explained in the section H,  How To Guide for Benchmarking a VLM (supplementary materials) as well as in the GitHub [2]. However, in order to further address this point upon critical reflection, we have added a ethical discussion on the dangers of performing both poorly and too well into the paper (see page 9 L327-344)

---

> > ### Author Response · Authors · 2023-08-17
> > **Response Part 2/2: Limitations, Correctness, Documentation, References**
> >
> > **Limitations (L)**
> >
> > **L1: While the authors claim that VisoGender can measure biases across multiple downstream tasks, the external validity of the scenes they curate and the natural language captions they construct is questionable**
> >
> > L R1: This work is a first attempt, in the VLM domain, to flag instances of (binary) gender bias in occupational settings. While we agree the scenes may be too constrained, these are the types of images we can reasonably expect to be used for image captioning and/or what will be retrieved in a search engine query. VISOGENDER is presented as a tool that could help detect a single level of bias, which is the difference between typical Western presentation of someone perceived as cisgender male vs cisgender female. Ability to efficiently check in this case, a level model’s bias towards heteronormativity presenting femininity in a workplace, could provide further opportunities to address other transgender and non-binary cases.
> >
> > **L2: The authors should reflect more critically on their decision to filter out occupations for which they could not find sufficient image data**
> >
> > **L R2:** We recognise this limitation, and hope that there is opportunity to further develop VisoGender to increase representation of occupations. As a preliminary benchmark, we use the most commonly depicted occupations and remain mindful that if a model still performs poorly and demonstrates biassed behaviour, it is likely to perform worse on occupations that are underrepresented.
> >
> > **Correctness (Corr.):**
> >
> > **Corr. 1: The benchmark is fundamentally problematic;**
> >
> > **Corr. R1:** We have removed reference to “ground truth gender” and reframed this as a “perceived gender label” which is further explained in the “Terminology used” section. Upon receiving the reviews, we reflected carefully on our language and terminology throughout the paper, and consulted with a group of domain experts and community members for their advice on revised terminology.
> >
> > Please see our response to Ethics Reviewer 2qKV “Issue 1”.
> >
> > Further, we have included a discussion on the Ethical Considerations in relation to this point. Please see subsection 5.2. On page 9, L327-344.
> >
> > **Corr. 2: Lines 300-302: The solution here is NOT to include non-binary people or neopronouns as additional categories; inferring gender and pronouns is problematic regardless**
> >
> > **Corr. R2:** Our benchmark is designed to measure occupational bias in VLM models, which we argue can lead to representational and allocational harms, as we cannot deny that gender biases and stereotypes exist and are embedded in existing models irrespective of the domain. We deem it prudent to include flexibility in our system, so that it can be adapted to include gender variation and neopronouns in future studies, as models have shown biassed performance with these pronouns, exacerbating the overall issue [3]
> >
> > **Corr. 3:  How do annotators guess the gender or pronoun of image subjects? This is not just a "complicated societal matter" (line 299); it is unethical.**
> >
> > **Corr. R3:** Please refer to Ethics Reviewer T9UW response: Concern 1.
> >
> > **Documentation (D):**
> >
> > **D 1: The authors provide clear background information and usage instructions on their GitHub. However, their repository does not contain a licence for their benchmark.**
> >
> > **D R1:** The License was previously stated in the README and in the Data Clause (Section J, Supp. Materials). However, in order to make this more clear, we have uploaded it as an additional file [4]. We have included terms of use, which specifically prohibit the use of the dataset for training. We hope the community will respect this and use the benchmark dataset for what it is intended, as a screening tool for flagging biassed models.
> >
> > **D 2: The authors do not discuss consent or copyright concerns regarding the images in their benchmark. In particular, image subjects may not consent to: 1) being included in the benchmark, and 2) their inferred gender annotations.**
> >
> > **D R2:** Our data clause (see supplementary materials and LICENCE in GitHub [4]) addresses privacy concerns . Please see the response 2 to Ethics Reviewer 2qKV. We have included a discussion point on this in subsection 5.2, page 10 L345-350. We have included a mechanism to allow anyone to opt-out, or request amendments to personal images [5]
> >
> > **References:**
> >
> > [1] https://arxiv.org/pdf/1804.09301v1.pdf
> >
> > [2] https://github.com/oxai/visogender#what-does-a-good-result-look-like
> >
> > [3] https://aclanthology.org/2023.acl-long.293.pdf
> >
> > [4] https://github.com/oxai/visogender/blob/main/LICENCE
> >
> > [5] https://github.com/oxai/visogender/tree/main#opting-out-removing-images-andor-adjusting-labels

---

> > > ### Comment · Reviewer_BTLm · 2023-08-18
> > > **Significant improvements to paper**
> > >
> > > I appreciate the authors' careful and thorough addressal of my concerns, as well as answering my questions; I have updated my score accordingly. In particular, the authors' updated treatment of gender; discussion of positionality; discussion of consent, privacy, maintenance, and data licensing concerns; and elaboration on ethical concerns significantly improve the quality of the paper. I am still uneasy about the methodology, especially the labeling of individuals' gender presentation.
> > >
> > > I would like to note that I understand and support the *motivation* behind VisoGender. To be clear, my issue was not exactly the lack of inclusion of "transgender, non-binary and gender-fluid" people, but rather that in its original form, the paper's treatment of gender (e.g., annotators inferring gender) was fundamentally problematic and incorrect; it has been improved in the updated version of the paper. I caution against language about "future work that extends the dataset to include underrepresented groups." We cannot keep extending our way to LGBTQIA+ inclusion when we do not begin with careful and reflexive considerations of gender. There has also been criticism of incremental approaches in technology development [1].
> > >
> > > [1] https://www.benzevgreen.com/wp-content/uploads/2019/11/19-ai4sg.pdf

---

### Official Review · Reviewer_Qbui · 2023-07-21
**The proposed dataset make significant contribution to measuring the societal bias of VLMs research area.**

**Rating:** 8
**Confidence:** 5

**Strengths:**

1. A novel dataset is created. The structure of the dataset and the image captions is well-documented.
2. The authors conducted experiments with evaluation of state-of-the-art models on obtained dataset.
3. The authors propose detailed ablation study.

**Additional Feedback:**

I suggest the authors to download the images and put it on some local storage, because the images could dissapear from third party resources and dataset becomes incomplete.



**Clarity:**

The paper is well written. There are some minor issues:
1. The paper contains several unnatural expressions, such as “a person in an occupation” (line 37) and “entrenching societal perceptions” (line 21).
2. I would also suggest using the phrase “social biases” instead of “societal biases”.


**Correctness:**

The evaluation methods and experiment design appropriate and performed correctly.

**Documentation:**

The authors proposed the repository with dataset, code to reproduce experiment results and documentation.

**Ethics:**

The authors do not specify who annotated the dataset: the crowdworkers or somebody from the authors. I suggest to add this information and provide wages if it was done by croudworkers.

**Limitations:**

The dataset limitations are addressed in detail

**Opportunities For Improvement:**

1. Selecting and labeling the images is discussed very briefly and the paper would benefit from a more structured description of these processes.
2. I suggest authors add error analysis. Is there any images, where all models provide wrong prediction?
3. Could it be that the model output is related to the colors in the picture? For example, a light picture is associated with a woman, and a dark one with a man, or something like that. It is easy to check with attention maps on images.
4. There is no explanation, why some models more accurately resolves men, and other women? Is it correlated with pre-trained dataset? Also BLIP-2 model has different type of bias on different tasks.

**Relation To Prior Work:**

The authors fully describe the relevant existing datasets and models

**Summary And Contributions:**

The paper presents VISOGENDER, an image dataset for testing occupational gender bias. The dataset contains 690 annotated images spread over 19 occupations and divided into two categories: “[a person of a certain occupation] and his/her [occupational tool]” and “[a person of a certain occupation] and his/her [client or otherwise associated individual]”. With the use of the dataset, several CLIP models and captioning models are tested on two tasks – determining the gender of the participant in an image and looking up relevant images for a gender-neutral search request – and behavior of two of the models (CLIP and BLIP-2) is investigated in further detail.

---

> ### Author Response · Authors · 2023-08-17
>
> We’d like to thank Reviewer Qbui for the feedback and review. It is validating to us to see our strengths recognised, and we appreciate the attention to detail. We have made the following changes in the paper, as well as provided responses to your questions.
>
> **Opportunities for improvement (OI)**
>
> **OI 1: Selecting and labelling the images is discussed very briefly and the paper would benefit from a more structured description of these processes.**
>
> **OI Response 1:** Please refer to the VISOGENDER Dataset Criteria in the Supp. Materials (see Section H). Further, please see the response from Ethics Reviewer T9UW Concern 1
>
> **OI 2: I suggest authors add error analysis. Is there any images, where all models provide wrong prediction?**
>
> **OI Response 2:** We have included an error analysis section in the supplementary materials for both resolution and retrieval bias (see Section F).
>
> *For the question: "Is there any images, where all models provide wrong prediction?"*, we have the following answer:
>
> 1)  CMEs had 27 images where all 6 models ((CLIP, DeCLIP, FILIP, OpenCLIP 400m, OpenCLIP 2b, SLIP) were wrong
> 2) Captioning models had 44 images where two models (BLIP v2, GIT) had incorrect responses
> 3) There were 4 images where all 8 models (both captioning and CMEs) were wrong. We double checked these images and confirmed they were definitely identifiable by humans.
>
> **OI 3:Could it be that the model output is related to the colours in the picture? For example, a light picture is associated with a woman, and a dark one with a man, or something like that. It is easy to check with attention maps on images.**
>
> **OI Response 3:** We agree that such analysis could be insightful, but is beyond the scope of our work. We advocate for future work to explore such spurious correlations models might have.
>
> **OI 4:There is no explanation, why some models more accurately resolves men, and other women? Is it correlated with pre-trained dataset? Also BLIP-2 model has different type of bias on different tasks**
>
> **OI Response 4:**
>
> We agree such analysis could be beneficial. There are some models in our analysis that are trained on similar data
> 1) BLIP-2 and OpenCLIP are trained on LAION (though BLIP-2 uses a 115M sample)
> 2) SLIP, DeCLIP and FILIP are trained on different subsets on YFCC100M
>
> However, we don’t see a clear pattern of the biases these models show that correlates with their training data.
>
> **Clarity (C):**
>
> **C1: The paper contains several unnatural expressions, such as “a person in an occupation” (line 37) and “entrenching societal perceptions” (line 21).**
>
> **C Response 1:** We have changed “a person in an occupation” to “depicted in an occupational role” (see page 2, L39). The phrase  “entrenching societal perceptions” was changed to “reinforcing stereotypical associations”. (see page 1, L24)
>
> **C2: I would also suggest using the phrase “social biases” instead of “societal biases”.**
>
> **C Response 2:** Thank you for the suggestion, we have made this change in the paper.
>
> **Ethics:**
>
> **The authors do not specify who annotated the dataset: the crowdworkers or somebody from the authors. I suggest to add this information and provide wages if it was done by croudworkers.**
>
> **Ethics response:** No work was carried out by crowdworkers, only by the authors. This is summarised in short in the main paper (Subsection 3.3, page 5, L186-191) then further detail is provided in Section H “VISOGENDER Dataset Criteria” in the Supp. Materials.
>
> **Additional Feedback:**
>
> **I suggest the authors to download the images and put it on some local storage, because the images could disappear from third party resources and dataset becomes incomplete**
>
> **AF Response:** We have decided not to download the images, as we do not have permission to host these images, and we are mindful that if they are removed by the parties hosting them, there may be reasons beyond our control.
>
> Therefore, we have included more detailed steps to our planned maintenance of the website. The authors undertake to check the dataset quarterly for any broken, or removed links and replace these with new images that abide by the correct licenced usage. This code is on Github as well, and requires a single command line instruction to assess the dataset, so that members of the community can assess the integrity of the URLs at any point. [1]
>
> We have also included an opt-out option on GitHub so that individuals can report problematic images of themselves, or any other individual. [2]
>
> **References:**
>
> [1] https://github.com/oxai/visogender#checking-the-data-integrity
>
> [2] https://github.com/oxai/visogender#opting-out-removing-images-andor-adjusting-labels

---

> > ### Comment · Reviewer_Qbui · 2023-08-30
> >
> > Dear authors, thank you for detailed comments and answers. I think this paper have a good opportunity to be accepted with current ratings and my rating is fair.

---

### Official Review · Reviewer_aXST · 2023-07-21
**Useful benchmark, needs to distinguish concepts related to gender**

**Rating:** 7
**Confidence:** 4

**Strengths:**

- **[Major] Useful adaptation of NLP metrics (WinoGender, WinoBias) to VLM models.** The adaptation seems well executed, provides a useful benchmark for certain kinds of VLM biases, and provides a interesting comparison point between LMs and VLMs.
- **[Major] Large benchmark of images for VLM evaluation.**
- **************[Minor]************** **Multiple useful bias metrics for VLMs — resolution and retrieval.** Benchmark enables multiple kinds of bias evaluation (perhaps even in addition to the metrics presented in this paper).
- **[Minor] Evidence of bias (particularly retrieval bias) in the performance of prominent VLMs.** This kind of evidence is socially significant and important for motivating future evaluation of VLM systems.

**Additional Feedback:**

How might these results relate to the training datasets for each model?

**Clarity:**

The paper is relatively easy to follow.

- **[Major] Distinguish between pronouns, gender identification, and gender presentation.** It might help to (early on) clearly delineate a person’s pronouns (the “language” side of things), gender presentation (the “visual” side of things), and gender identification (related to both). The benchmark labels are based on gender presentation, but the resolution metrics are about pronouns, and the retrieval task (as well as the [dataset labels](https://github.com/oxai/visogender/blob/main/data/visogender_data/OO/OO_Visogender_11062023.tsv), table captions and other verbiage) often refer to “gender” (when really I think they refer to gender *presentation*, as ascribed to images by the annotators). Also, people with she/her pronouns are not necessarily “female” and he/him pronouns are not necessarily “male” (amend L291-292). These are all different concepts, and confusing them makes it hard to make sense of the results. Defining these terms early on and using them consistently and correctly could help with the points above — part of the importance of this study is that VLMs contain biases related to both visual and linguistic aspects of gender, and the paper would do well to emphasize that nuance.
- **[Minor] Mention which metric is being plotted in the Fig. 3 caption.** I had to check the text to see if it was resolution or retrieval. Also, you might mention the correlation metrics for the retrieval bias metrics as well — I see from the appendix Fig. 6 that they are positive but not as large, which is interesting.

**Correctness:**

The analysis looks correct and the methods appropriate, except for the lack of error/validity metrics as noted above (although I acknowledge that including error bars or other error metrics is not the norm in this kind of paper).

**Documentation:**

The Datasheet is pretty good, but the Checklist is missing some details.

- 3c — Why not report error bars?
- 4b — The Datasheet details that the authors only used Creative Commons licensed images, but it isn’t mentioned in the main text. Also, which CC license terms were permitted? Does the dataset include non-commercial or attribution licenses? Does the dataset include attribution?
- 4d — The main paper doesn’t mention consent. The datasheet mentions that the authors assumed that CC licensed images received consent from the subjects — see notes below.

**Ethics:**

The Datasheet mentions that the authors assumed that CC licensed images received informed consent from the subjects. What’s the basis for this assumption? As far as I know, CC licenses do not require the creators to have gotten permission from subjects, so this assumption seems unfounded.

Since this dataset is for evaluation, not training, the images are not NSFW and of adults, and the dataset only includes URLs, I don’t think is a huge ethical issue, but I’d suggest:

- Offering at least a simple opt-out option on the Github page, in case anyone does want their image URL removed.
- Placing the dataset under a license that specifically prohibits uses other than model evaluation, since the authors note in the Datasheet that this dataset is only intended for evaluation.

**Limitations:**

The authors do well to reflexively acknowledge their subjectivity in inferring pronouns for the benchmark, but that reflexivity should extend to how the results are framed and interpreted, as suggested above.

**Opportunities For Improvement:**

I think this benchmark is flawed (primarily because it relies on inferred gender labels) but useful. An excellent paper would address the issues below, but I think the paper can be accepted so long as the authors make some additional disclaimers and further clarify some limitations of their results (below).

- **********************************[Major] Remove references to “groundtruth” labels and reframe results as comparisons to stereotypical human pronoun resolution.********************************** Throughout the papers, the authors refer to models’ ability to “correctly” infer (L53) “groundtruth” (L98) pronouns from images. In L75, the authors claim their benchmark is different from Winograd in that “the correct resolution is unambiguous, i.e. there is a correct caption (and pronoun) for a corresponding image.” But as the authors acknowledge in the conclusion, the pronouns assigned to images in the evaluation dataset are by no means “correct” or “ground truth” — they are assigned by the authors, not self-identified. What if the authors have inadvertently misgendered (here I use misgendered to mean mistaken pronoun assignment, though gender and pronoun preference are not 1:1) one or more of the images in the dataset (particularly for images of trans or nonbinary individuals)? With so many images, it seems quite possible. A truly outstanding benchmark would include self-identified pronoun/gender markers, the “correct” labeling — and my score would increase greatly if the authors built a benchmark with this data, through a stock photo company or a new IRB-approved study. I understand the practical limitations of collecting images with self-identified pronouns or gender, and the dataset the authors put together is still valuable — however, it seems misleading to describe the pronoun labels as “groundtruth.” As the authors acknowledge, those labels are subjective (subject to both the authors individual biases and the binary construction of gender inherent in stock images, search engines, and the internet at large). It would be more accurate to say that the results in this study compare the performance of the VLMs to the typical inferences of (human) researchers, who are also prone to misgendering — using language like “ground truth” obscures this fact.
- **[Major] Retrieval benchmark encodes stereotypical gender presentation.** It seems like VLMs could have two kinds of bias that affect performance on pronoun resolution: stereotyping, i.e. disproportionate mistakes in assigning pronouns to occupations; and misgendering, i.e. misinterpretation of physical gender presentation into pronouns (after controlling for occupational contexts). For example, it’s possibly (even likely) the case that a cis woman with she/her pronouns wearing a hard hat and a trans woman with she/her pronouns wearing a hard hat could receive much different treatment from the model. The intro mentions both, but this paper really only seems to measure the former (heteronormative stereotyping) — in fact, the benchmark potentially *encodes* misgendering bias, since the authors are inferring (and potentially misgendering) the pronouns of the subjects. The models could perform perfectly well on stereotypically male- and female-presenting stock models, but fail miserably on people who don’t appear stereotypically male- or female-, or people whose pronouns and gender identity don’t match their gender presentation. Ideally, the authors could construct a benchmark with a) self-identified pronouns/gender, and b) diverse pronouns / gender identity, and include a baseline test for misgendering (e.g. the single person test, but without any occupational context; if the results in Tab. 2 hold, the incidence of non-occupational misgendering might be quite high already). At the least, they should emphasize that 1) these results probably only hold for cisgendered individuals who present stereotypically, 2) the models may perform quite differently in relation to people’s actual pronouns/gender (e.g. if the models misgender people in the same way as the annotators, their disparities could be worse than these results indicate), and 3) some harms could be avoided by avoiding gendering entirely (as the authors mention in L263, CLIP, for example, may cause less harm if it is significantly more likely to use neutral pronouns).
- **[Minor] Articulate the potential applications of this bias that could cause harm.** The first paragraph has a couple good examples, but the rest of the paper tends to refer only to disproportionate accuracy or other bias metrics. When defining the metrics, also mention the kinds of harms that could arise (search engine representational stereotypes, misgendering, etc.). Also, how important is pronoun resolution? How similar is the template used for evaluation to e.g. open-ended captioning?
- **********[Minor] Report sample sizes / confidence intervals / significance alongside bias metrics.********** How statistically different are the metrics reported in Tables 1-4? Can you provide any error bars, confidence intervals, effect sizes, sample sizes, or any other info to help readers understand how different these effects are from each other and from zero?

**Relation To Prior Work:**

Yes. The authors might also consider mentioning work on text-to-image systems (e.g. [Bianchi et al., 2023](https://dl-acm-org.cmu.idm.oclc.org/doi/abs/10.1145/3593013.3594095) or [Cho et al., 2022](https://arxiv.org/abs/2202.04053)), if they find it relevant.

**Summary And Contributions:**

The authors compile a dataset of images for evaluating VLM biases, including pronoun resolution and gendered image retrieval, and present evidence that prominent captioning and image retrieval models perpetuate certain occupational gender biases.

---

> ### Author Response · Authors · 2023-08-17
>
> We thank Reviewer aXST for the feedback which we have used to improve the terminology and aims within VISOGENDER
>
> **Opportunities for improvement (OI):**
>
> **OI 1: [Major] Remove references to “groundtruth” labels**
>
> **OI R1:**
> Thank you for the comment. We have done the following:
> 1) We have reworked our use of terminology throughout the revised paper to remove references to a groundtruth, and included a “Terminology used” on page 4, subsection 3.1.
> 2) We have added an option (in addition to opting out) that allows individuals that identify themselves in the images to request the changing of their labels, as necessary. This is set up with a Google Form on the repository which send notifications to the first author each time an entry is made. [1]
>
> **OI 2: [Major] Retrieval benchmark encodes stereotypical gender presentation**
>
> **OI R2:** Thank you for this observation, it has helped us shape our narrative and we have taken steps to ensure this clarification is made. We have added statements to emphasise that this benchmark is based on a Western-centric heteronormative view of binary (cis)gender (for example, see introduction pages 2-3 L69-71 ). We acknowledge our encoded biases in the limitations section, with the inclusion of a positionality statement.
>
> **OI 3:[Minor] Articulate the potential applications of this bias that could cause harm**
>
> **OI R3:** Please see our ethical considerations section for a more detailed assessment. Pronoun resolution provides us a tool to evaluate how much a VLM associates subjects with certain gender presentations with certain occupations. For example, if an automatic captioning VLM in a downstream application repeatedly misgenders doctors as “he”, this erases the representation of “women'' as doctors. By extension, this misgendering is likely to affect LGBTQIA+ representation as well. Please see Section 3.4. for the template description
>
> **OI 4: [Minor] Report sample sizes**
>
> **OI R4:** We have included an error analysis (Section F in the Supp. materials). For a summary of the results, please see the response “OI3” to Reviewer 6G4g
>
> **Clarity (C):**
>
> **C1:[Major] Distinguish between pronouns, gender identification, and gender presentation**
>
> **CR 1:** Please see our new terminology (See Page 4 subsection 3.1) in the revised paper
>
> **C2: [Minor] Mention which metric is being plotted in the Fig. 3 caption**
>
> **C R2:** This is resolution bias, and we have since updated this (please see the light purple highlight for Fig. 3).
>
> **Documentation (D)**
>
> **D1: 3c — Why not report error bars?**
>
> **DR1:** We have included an error analysis (please see Sec. F in the Supplementary materials) and we have updated the checklist.
>
> **D2:  4b - CC licences, attribution etc.**
>
> **D R2:** We now make reference to the licences in the text (see page 5 in the footnote). The licence is CC-BY 4.0 and we have included a Terms of Use (see Supp. Material subsection K1) which limits the dataset to be used as a benchmark only. Individual images in the dataset can have non-commercial licence use, but VISOGENDER is not in violation of this. Anyone using the URLs outside of the terms of use we stipulate will be liable to those licence constraints. With regards to attribution, according to the CC guidelines, there’s no right or wrong way to attribute the credits to a work and the attribution must be done in a ‘reasonable’ way. Within the scope of the benchmark where only link to URLs and record licences we believe this obligation is met within reason.
>
> **D3: 4d - Issue of consent.**
>
> **D R3:** Please see the response to Issue 2 for  Ethics Reviewer 2qKV
>
> **Ethics (E)**
>
> **E1:  The Datasheet mentions that the authors assumed that CC licensed images received informed consent from the subjects**
>
> **ER1:** We were following the style of previous papers such as Gender Shades [2] and the Adience Dataset [3], however we recognise that our description is not accurate. We are not able to obtain informed consent of the assigned perceived gender labels, but we use the images in VISOGENDER in line with even the most restrictive CC licences (that is, we don’t use it for commercial purposes, nor make any derivatives). Please see Ethics Reviewer T9UW Concerns 1 & 4 for more details
>
> **E2: … dataset is for evaluation, not training, the images are NSFW and of adults ...**
>
> **ER2:** Thank you for these suggestions, we have done both of these. There is a Google Form which will be monitored [1], and allows for opting out or adjusting of any labels. We have amended our licence to be more explicit in what it is used for, i.e. only training. Please note, **our images are safe-for-work**. We have added a clarification about this to the Supp. Material section F on page 3.
>
> **References**
>
> [1]  https://github.com/oxai/visogender#opting-out-removing-images-andor-adjusting-labels
>
> [2] http://proceedings.mlr.press/v81/buolamwini18a/buolamwini18a.pdf
>
> [3] https://ieeexplore.ieee.org/stamp/stamp.jsp?tp=&arnumber=6906255&tag=1

---

> > ### Comment · Reviewer_aXST · 2023-08-29
> > **Good improvements**
> >
> > Thanks to the authors for their diligent responses and the extensive improvements. The language is much better now, in my view --- it is easier to see exactly what assumptions about gender are being made in the methods, and to understand how those assumptions might affect the results. I agree with other reviewers that this approach to benchmarking is flawed, but I also agree with other reviewers that there is still value in this kind of approach as long as the downsides are clearly stated and acknowledged, as they are now.
> >
> > I've increased my score accordingly.
> >
> > P.S. Apologies, "NSFW" was a typo --- I meant "not NSFW". Thanks anyways for the clarification.

---

### Official Review · Reviewer_6G4g · 2023-07-23
**Well-designed dataset but needs work on the model benchmarks and results presentation**

**Rating:** 7
**Confidence:** 4

**Strengths:**

Uncovering implicit gender biases in vision-language models is very consequential for many downstream tasks such as retrieving potentially useful stock images. Models' inherent biases can therefore contribute to perpetuating harmful gender stereotypes. The proposed dataset and evaluation pipeline can help identify such risks. The dataset construction is well controlled and the evaluation methods appear carefully designed. The discussion of the limitations is very thoughtful. A thoroughly completed datasheet is attached with the supplementary materials.

**Additional Feedback:**

The conclusion is tying together the paper very nicely.

**Clarity:**

Overall, the paper is well-written but especially the results section can be made clearer. I'm listing some suggestions here but they don't have to be integrated. They're just meant as an inspiration that might make that section easier to follow.

1: The distinction between accuracy and bias could be elevated to help highlight which results are actually about bias detection and which are about model capabilities.

2: I recommend reducing the space taken up by figures and tables (potentially even moving some more to the appendix, e.g., figure 3 or many of the different metrics in Table 4 since they're currently not discussed in the text anyways) to make more space for interpretation of the results. Currently, this section primarily lists results and refers to tables and figures but the lack of take-aways makes it hard to keep track of the narrative. Section 4.3 for instance seems interesting and quite crucial to the main narrative since it's the primary result for one of the two tasks but there is no interpretation of the results of why the two metrics might give so contrasting results and what that means for the benchmark.

3: Figure 2, for example, has 2 subplots but they're not distinguished when referenced in the text. Labeling and referencing subfigures can help the reader focus their attention on the crucial data.

4: I recommend restructuring the subsections. The main tasks are resolution and retrieval. For both of them, it seems there are accuracy and bias results. The next section is a discussion section. Maybe the following structure could work: 4 Results; 4.1 Resolution Task; 4.1.1 Accuracy; 4.1.2 Bias; 4.2 Retrieval Task; 4.2.1 Accuracy; 4.2.2 Bias. Then 5 Discussion; 5.1 Ablations; etc. This way, the central findings are all in one section without distractions.

**Correctness:**

The dataset appears to be constructed in a sound way. The evaluation methods and experiment design also appear well-designed but I can only fully judge that once my questions about the model comparisons are addressed.

**Documentation:**

Yes.

**Ethics:**

No.

**Limitations:**

Yes.

**Opportunities For Improvement:**

Overall, the paper is well written, but the presentation of the results is a little hard to follow and I recommend streamlining it more. I added suggestions for that in the "Clarity" section. Other Opportunities for Improvement largely relate to what I think might be misunderstandings since the presentation of the results was slightly unclear to me but I believe they warrant addressing.

The authors introduce a CLIP vs. captioning model distinction, but I couldn't quite follow the nature of that distinction. Based on section 3.3, I'm assuming that it's a distinction between models that were trained with a contrastive learning objective (like CLIP) and models capable of autoregressively generating image-based text (like BLIP). However, I'm unclear whether these categories are intended to be mutually exclusive -- where would ClipCap (Mokady et al. 2021) go? Due to the naming of the first category as "CLIP models", the authors currently categorize FILIP (Yao et al. 2021) as a CLIP model which doesn't seem correct since FILIP doesn't contain any actual CLIP model components (as per my understanding). What connects them is the training objective. For that reason, the model categorization throughout the paper is very confusing and I recommend refining the definition of the model groups that are being compared.

Relatedly, the two groups of models often appear to be compared against each other, but mainly just by using the original CLIP model and the BLIP model as representatives. However, this requires that the paper establishes that models within these categories behave alike and that CLIP and BLIP are actually representing those groups. If it's simply a model comparison and CLIP and BLIP are arbitrarily chosen, this also needs to be highlighted.

Many effects discussed in the paper (Tables 2, 3; Figures 2, 4, 5) seem subtle which makes me worried how many of the results still hold when error bars over multiple runs or model variants are added. Any potential estimate of robustness will help support the conclusions.

I think the BLIP-2 results in Table 3 need to be clarified. Currently, the paper states that "From Tab. 3 we see that models tend to exhibit a larger resolution accuracy gap as we go to more “difficult” subtasks, such as two people with different genders where there is higher variation and almost random predictions across models." However, the BLIP-2 results remain at approximately 0 across tasks. This seems to be at odds with the narrative.

**Relation To Prior Work:**

Yes.

**Summary And Contributions:**

The authors propose a dataset to benchmark gender bias in vision-language models, focusing on occupational biases. The dataset consists of images paired with referential clauses such as "the doctor and his/her patient".  Bias is measured based on two tasks: (pronoun) resolution and (image) retrieval. The authors benchmark a variety of CLIP-based models and BLIP, and find that gender biases are especially pronounced when there are two people in the images with opposite genders. For BLIP but not CLIP, these biases are aligned with the bias in the representation of those genders in the US.

[overall rating updated after rebuttal]

---

> ### Author Response · Authors · 2023-08-17
>
> Thank you to Reviewer 6G4g for the feedback. The review helped us reflect on the organisation of the paper. We have taken the majority of this feedback into account, and believe the message of the paper is clearer as a result.
>
> **Responses:**
>
> **Opportunities for improvement (OI)**
>
> **OI 1:  CLIP vs. captioning model distinction:**
>
> **OI Response 1:** As we discuss in L168-169 (original manuscript), we define ‘CLIP models’ as joint vision and text encoders that output a similarity score between images and text, but make no claims about the losses they are trained with (they don’t have to be contrastive), or their specific components. ClipCap (Mokady et al. 2021) outputs a caption, and not a similarity score, and as per our definition, is to be analysed as a captioning model. We agree that ‘CLIP models’ is confusing as there are more such models, and have renamed this family of models to ‘Cross Modal Encoders’ (CMEs), using this term throughout the revised paper.
>
> **OI 2:  The use of CLIP and captioning models as representatives:**
>
> **OI Response 2:** We compare high level statistics on several models from both classes. When we evaluate in further detail, we focus on CLIP and BLIP, but these are not arbitrarily chosen –  they are among the most widely used models from their respective family (as seen from total model downloads on Huggingface). We have included this motivation to the revised paper. Please see Section 4, page 7, L271-273.
>
> **OI 3:  Any potential estimate of robustness will help support the conclusions:**
>
> **OI Response 3:**  We’d like to refer the reviewer to the newly added section F in the appendix titled “Error Analysis” where we run these tests. In short, we find the following:
> 1) Resolution bias: we demonstrate that sub-sampling our dataset does not skew the results by much on average. Please see Figure 12 and Table 8, both on page 27 for more details.
> 2) Retrieval Bias: we find that the models are more biassed when split by perceived gender presentation, as opposed to randomly splitting the dataset. Please refer to Table 9, page 28 for more details.
>
> **OI 4:  Clarifying the BLIP-2 results in table 3:**
> **OI Response4:** Apologies, this is an oversight that we have rectified (see light purple highlight on page 8 L289). This should have referred to Table 2.
>
> **Clarity (C):**
>
> **C1:  The distinction between accuracy and bias could be elevated to help highlight which results are actually about bias detection and which are about model capabilities.**
>
> **C Response 1:** We thank the reviewer for the remark. We have updated subsection 4.1. Page 7 L278-279.
>
> **C2:  Presentation of the results:**
> **C Response 2:** We thank you for the suggestion and this was incorporated with other feedback in the reviews. This is a summary of the changes:
> *Results:*
> 1. Moved the ablations for resolution bias to the supplementary materials
> 2. Moved Figure 5 (Retrieval bias) to the appendix where occupation results are discussed
> 3. Changed reference of CLIP models to ‘Cross Modal Encoders’ (CMEs).
>
> **C4:  Restructuring the subsections:**
>
> **C Response 4:** We thank you for the suggestion. Upon reflection, taking into account the addition of the Ethical Considerations and Positionality Statement to the Discussion section, it is clearer to remove the ablations from the main paper entirely and only focus on the core benchmark results.  Please see “Section C “Ablations” in the Supplementary Materials.

---

> > ### Comment · Reviewer_6G4g · 2023-08-18
> >
> > I thank the reviewers for their thorough response. I've updated my overall recommendation.

---

### Official Review · Reviewer_EykH · 2023-07-24

**Rating:** 7
**Confidence:** 3

**Strengths:**

This paper has many strengths:
* The motivation for the work is strong and compelling
* This dataset tackles a persisting communal need for well-annotated datasets to evaluate model bias
* The proposed tasks are straight forward, which may ease community adoption
* The benchmarks are recent and relevant and preliminary findings are interesting

**Additional Feedback:**

- Line 135: "occuptation"
- Line 139: "two subjects but assigning" -> "two subjects by assigning"?
- Line 144: "and inferring" -> "and infer"?
- It's confusing to add "ZS Imagenet" as another column in the Resolution Accuracy Table (2) because it appears to be a different metric.
- It's not obvious how we can tell where the gaps come from different man/woman distributions or from "bias" in the model... Since deltas are small and dataset size is small, it would be helpful to ensure gaps arise from gender bias, not variance due to sample size, for example. Perhaps another evaluation could be run: Randomly split your data into groups NOT by gender and report avg. differences to have a distribution of gaps to compare against? Maybe a gap of 0.2 is massive because when not splitting by gender it's usually near 0.

**Clarity:**

The paper is overall written well (some suggestions in "Additional Feedback" below)

**Correctness:**

More details of data collection could be provided, but the dataset appears sound and experiments appropriate.

**Documentation:**

Details on data collection could be added to the paper, the dataset repository looks clean and complete.

**Limitations:**

The authors indeed list 3 key limitations, which is sufficient.

**Opportunities For Improvement:**

This paper could be improved with the following considerations:
* The dataset collection process could be described in more detail: What steps were taken to ensure images are semantically similar between genders? How were image sources selected and what downstream tasks might these align with?
* Some of the experiments could be explained in further detail. For instance, the gaps look quite low (down to -1) in Figure 3 but the tables report much smaller gaps. Why are these numbers so different?
* It would ultimately be helpful to also show some applications of mitigation strategies on top of benchmarking. Is there a way these data could eventually be integrated into the training process to lead to better models?

**Relation To Prior Work:**

The authors discuss differences from prior works.

**Summary And Contributions:**

This paper introduces a new dataset that can be used to evaluate gender bias in vision-language (VL) models. The authors collect 690 images containing different numbers of people and different genders. Then, given their gender annotations for people in each image, they describe two tasks on which they evaluate VLs. First, they propose *pronoun resolution*: Correctly identify the gender pronouns as either "he/his" or "she/her". Second they propose *gender retrieval*: Given a gender-neutral caption, report the gender associated with people in the retrieved images. The authors then contribute an evaluation of recent VL models on these tasks and find that gaps in performance indeed exist on these tasks. For the resolution task, large gaps exist when identifying pronouns in images of two people with different genders. Interestingly, the models have a propensity to retrieve pronouns associated with women than men, though the gaps do get smaller for the best models overall. For the retrieval task, the compared models are much more likely to retrieve images of men.

---

> ### Author Response · Authors · 2023-08-17
>
> We’d like to thank Reviewer EykH24 for the helpful feedback and interesting questions. Please find our responses below.
>
> **Responses:**
>
> **Q1: Describing the data collection in more detail**
> **A1:** Image selection and labelling was done by the authors. Due to funding constraints, and an inability to pay fair wages for crowdworkers, we decided to keep data collection internal. We acknowledge the tradeoff with the potential bias this introduced in the positionality statement, added to the limitations (See Sec 5.3.). The image sources are described in supplementary materials Section F “VISOGENDER Dataset criteria”. When selecting images, we selected scenes that were stereotypical of the profession (i.e. the positioning of the people, their environment) and chose people depicted in an occupational role that had stereotypical clothing. The participant was typically chosen if they were in casual clothing. In order to ensure some diversity in age and race, these pictures selected came from slightly different sources, reducing the likelihood of spurious correlations.
>
> Image sources were selected based on their associated licences. We chose platforms that had Creative Commons licences and/or royalty free images that did not prohibit the use of the images for machine learning. These platforms would align with downstream tasks such a typical image search query through a search engine, as well as the automatic generation of alt-text on an image-sharing platform.
>
> **Q2: Some of the experiments could be explained in further detail. For instance, the gaps look quite low (down to -1) in Figure 3 but the tables report much smaller gaps. Why are these numbers so different?**
>
> **A2:** Figure 3 shows per occupation gaps, whereas the tables in the paper show average gaps, across all occupations. For more detailed per-occupation results, please refer to Appendix B the Supplementary Material. We have updated the caption to make this more immediately clear.
>
> **Q3: It would ultimately be helpful to also show some applications of mitigation strategies on top of benchmarking.**
>
> **A3:** This paper is aimed at introducing a benchmark, so that AI researchers can have an evaluative guideline for evaluating how well their bias mitigation strategies work. We believe bias mitigation strategies are beyond the scope of this paper and therefore make no attempt to include them.
>
>
> **Documentation: Details on data collection could be added to the paper**
>
> **Response:** Please see the response to Ethics Reviewer T9UW Concern 1 .
>
> **Additional feedback**
> Thank you for your attention to detail - we have corrected the typos.
>
> **AF 1: It's confusing to add "ZS Imagenet" as another column in the Resolution Accuracy Table (2) because it appears to be a different metric.**
>
> **A1:** A model that makes random predictions would have a zero accuracy gap and retrieval bias (as it makes random predictions and thus can not be biassed). Thus we advocate including zero-shot Imagenet classification accuracy in bias results tables.
>
> **AF2: Addressing robustness concerns**
>
> **A2:** We’d like to refer the reviewer to the newly added section F in the appendix titled “Error Analysis” where we run these tests. In short, we find the following:
>
> 1) Resolution bias: we demonstrate that sub-sampling our dataset does not skew the results by much on average. Please see Figure 12 and Table 8, both on page 27 for more details.
> 2) Retrieval Bias: we find that the models are more biassed when split by perceived gender presentation, as opposed to randomly splitting the dataset. Please refer to Table 9, page 28 for more details.

---

> > ### Comment · Reviewer_EykH · 2023-08-29
> >
> > Thank you for your detailed responses. I'm keeping my score the same since I already think this paper passes the bar for acceptance.
> >
> > A few notes:
> > * Reviewers don't seem to get access to responses to ethics reviews, so I can't vouch for your answers there.
> > * I do share some of the concerns mentioned in the ethics reviews surrounding binary gender/pronouns. While addressing this in writing is good, I'm not convinced that's really enough: since benchmarks like this can get used many times, the rationale for this decision can be washed away, perpetuating erasure in follow-up evaluations. I don't think that's enough to bar acceptance, but I do think it's enough to warrant concerns about the impacts of this work. To think from the opposite direction: A more-representative benchmark would be that much more impactful and beneficial!

---

### Author Response · Authors · 2023-08-17
**General Comment to Reviewers, Ethics Reviewers, and Area Chairs**

We thank the reviewers and the ethics reviewers for their valuable feedback. We are encouraged that our benchmark was considered valuable for evaluating model bias in vision-language models.

We especially thank the reviewers for such detailed and rigorous reviews. We’d like to emphasise that we take the ethical concerns about data privacy, consent and the adjacent issues of automatic gender recognition technology very seriously. Further concerns were raised about our conflation of pronouns, gender presentation, and gender identity and the lack of clarity around labelling. The majority of additions to the paper (outlined below) are in response to these points.

In developing this benchmark, we acknowledge that we have not included all underrepresented communities, and we especially recognise our limitations in addressing LGBTQIA+ communities, specifically non-binary gender representations. However, we believe VISOGENDER remains an important benchmark, as it provides a means for evaluating and flagging when some forms of binary gender bias may be still present. This is the first benchmark of its kind for VLMs, and builds on important work in NLP.  We hope this benchmark inspires future work that extends the dataset to include underrepresented groups, especially those identifying as transgender, non-binary and gender-fluid as well as more diverse occupational roles that we were not able to provide.

Further, we recognise that a form of pronoun resolution is included and would like to stress that the aim of the benchmark is not to predict the gender identity of anyone in the images. In the revised paper, we aimed to carefully discuss and reflect on how perceived gender and inferred pronouns may lead to misrepresentation, as well reinforce binary, stereotypical and Western-centric systems of gender presentation
All revisions to the text are highlighted in light purple in the manuscript. If an entirely new section is added, we have only highlighted the heading. If this colour is unsuitable, please let us know and we will revise it.

Summary of changes:

1. Abstract:
 - We have updated our terminology based on the points raised below.
2. Introduction:
- A reframing of the introduction to acknowledge that our work relates to a Western-centric view of heteronormative gender, while acknowledging that it is likely that models will perform even worse in non-stereotypical cases with underrepresented groups where there will be less training data.
- We introduce the idea that performing well on this benchmark requires not only advanced visual-linguistic reasoning but also accurate gender prediction, a capability that is concerning if misused for surveillance purposes. We raise this ethical concern in Section 5 in more detail.
3. VISOGENDER Section
- Inclusion of  a terminology subsection to Section 3,
- Removal of references to a “groundtruth label” in the manuscript and the code. This is now described as a perceived gender label.
- We no longer refer to “men” and “women” (except for the exceptional case of US statistics) and make reference to perceived gender presentations (masculine and feminine).
4. Results:
- Moved the ablations for resolution bias to the supplementary materials
- Moved Figure 5 (Retrieval bias) to the appendix where occupation results are discussed
- Changed reference of CLIP models to ‘Cross Modal Encoders’ (CMEs).
5. Discussion Section:
- Inclusion of an ethical considerations section:
- Discussion of harms of performing both poorly and well on this dataset (see page 9, subsection 5.2)  to address the automatic gender recognition risk
- Discussion of privacy and data licencing concerns
- Inclusion of our positionality statement to the limitations
6. Supplementary materials
- We have updated the data clause outlining the licence, terms of use, a detailed maintenance plan and the option for anyone to request images be removed [1] and/or any adjustments be made to assigned labels. This is updated on GitHub as well [2].
- Datasheet: this has been updated to address privacy concerns, the issues of consent,  the maintenance plan, a mechanism to report errors and/or opt-out of any images.
- Multiple questions were raised about the lack of error bars / robustness. In order to address this, we have included an error analysis section in the supplementary materials for both resolution and retrieval bias (see Section F)
7. GitHub
- Code and dataset have been updated to match terminology
- An opt-out mechanism has been included [2]
- Licence is in its own file now [3]

**References:**

[1] https://github.com/oxai/visogender/

[2] https://github.com/oxai/visogender#opting-out-removing-images-andor-adjusting-labels

[3] https://github.com/oxai/visogender/blob/main/LICENCE

---

### Decision · Program_Chairs · 2023-09-22

**Decision:**

Accept (Poster)

**Comment:**

This paper presents the WinoGender dataset, which is designed to test biases in multimodal models.  This dataset is very timely and will likely spur thought in the space of multimodal bias, an important area of study.  I appreciate the efforts from the authors to respond to reviewer comments and believe this will make a nice addition to the conference.